# Composite SMG5-SMG6 PIN domain formation is essential for NMD

Katharina Kurscheidt [1,5], Sophie Theunissen [2,3,5], Natalia Pasquali[4], Kerstin Becker [4], Volker Boehm [2,3], Elena Conti [1] ✉ & Niels H. Gehring [2,3] ✉

Nonsense-mediated mRNA decay (NMD) relies on the coordinated assembly and action of multiple protein factors. Degradation of target mRNAs begins with endonucleolytic cleavage near premature stop codons, but the mechanisms of endonuclease activation and regulation remain unclear. Using structural predictions, biochemical in vitro assays, and cell-based NMD analysis, we show that SMG5 and SMG6 interact via their PIN domains to form a composite interface (cPIN) with full endonuclease activity. In vitro reconstituted SMG5-SMG6 cPIN heterodimers show high activity, as SMG5 completes the SMG6 active site and substrate binding site. Mutations in residues at their predicted interaction surfaces, RNA-binding sites, or active site attenuate or abolish cPIN activity in vitro and impair cellular NMD. Our findings demonstrate how paralogous PIN domains complement each other to assemble a highly active endonuclease in NMD, providing a structural and mechanistic explanation for efficient NMD substrate degradation.

Nonsense-mediated mRNA decay (NMD) is a eukaryotic surveillance mechanism that eliminates transcripts harboring premature termination codons (PTCs)[1]. Thereby, NMD prevents the synthesis of potentially deleterious truncated proteins and fine-tunes the expression of a substantial portion of the transcriptome[2]. NMD is triggered by premature termination of a ribosome, creating an aberrant mRNP architecture that is recognized by the helicase UPF1[3]. Subsequent phosphorylation of UPF1 by the kinase SMG1 initiates the recruitment of the effectors SMG5, SMG6 and SMG7 and ultimately commits the transcript to degradation[4,5]. This recruitment marks the transition from NMD target recognition to transcript degradation, yet the precise roles of SMG5, SMG6, and SMG7 remain unclear.

SMG5 and SMG6 are essential NMD factors across metazoans, including nematodes[6], flies[7] and mammals[8,9]. Both proteins share a broadly similar domain organization: an N-terminal (SMG5) or central (SMG6) tetratricopeptide repeat-like (TPR-like) helical domain with a 14-3-3-like fold followed by a helical-hairpins domain, and a C-terminal PilT N-terminus (PIN)-like domain[4,10]. Despite this architectural similarity, key differences are well established. The 14-3-3-like folds of SMG5 and SMG6 are similar in that neither can bind phosphorylated UPF1, unlike the corresponding domain in SMG7[11]. However, they differ in their oligomeric states within the course of NMD: in SMG5, the 14-3-3-like fold forms a heterodimer with the 14-3-3-like domain of SMG7[10], whereas SMG6 remains monomeric[4]. In both proteins, the TPR-like helical domains include extended loop regions and are connected via a long α-helix to the C-terminal domains, which contain a PIN-like fold[4,10]. The key difference between the PIN domains of SMG6 and SMG5 is that SMG5 lacks the catalytic residues and, consequently, the endonuclease activity present in SMG6[12–15]. Although structural studies have revealed the architecture of individual domains, their cooperation within functional NMD assemblies remains unresolved. Consequently, the molecular basis by which SMG5 and SMG6 domains act together during transcript degradation is still unclear.

Two major NMD branches have been established in metazoans, a SMG6-dependent endonucleolytic pathway and a SMG5-SMG7-mediated deadenylation/decapping pathway, where the SMG5-SMG7

[1]Department of Structural Cell Biology, Max Planck Institute of Biochemistry, Martinsried, Germany. [2]Institute for Genetics, University of Cologne, Cologne, Germany. [3]Center for Molecular Medicine Cologne (CMMC), University of Cologne, Cologne, Germany. [4]Cologne Center for Genomics (CCG), Medical Faculty, University of Cologne, Cologne, Germany. [5]These authors contributed equally: Katharina Kurscheidt, Sophie Theunissen. ✉e-mail: conti@biochem.mpg.de; ngehring@uni-koeln.de

heterodimer acts primarily as a connector between phospho-UPF1 and general decay machineries[4,10,16]. Paradoxically, the SMG5 PIN domain is required in cells for SMG6-mediated cleavage[17], even though SMG5 lacks nuclease activity of its own[13]. However, the proposed model of two independent and partially redundant branches in the NMD pathway in metazoans does not explain the weak activity of SMG6 in isolation in vitro, the requirement for the SMG5 PIN domain in SMG6-dependent cleavage, and the apparent absence of a stable SMG5-SMG6 complex in cells[13,14,17,18].

Here we address this mechanistic gap by combining structure prediction with in vitro nuclease assays and NMD rescue assays in cultured cells. Guided by AlphaFold-based modeling, we identify a direct PIN-PIN interface between SMG5 and SMG6 and confirm its functional relevance with purified proteins and targeted mutagenesis. These analyses reveal the molecular mechanisms with which the SMG5 PIN domain directly stimulates SMG6 activity, namely by completing composite catalytic and RNA-binding sites. Mutations that disrupt the predicted RNA-binding patch or the PIN-PIN interface abolish SMG5-dependent stimulation in vitro and impair NMD in cells. Overall, this explains how recruitment pathways that are differentially sensitive to UPF1 phosphorylation converge to activate endonucleolytic cleavage during human NMD.

## Results

### Direct SMG5 - SMG6 interaction stimulates endonucleolytic activity in vitro

Recent evidence indicates that the SMG5 PIN domain is required for SMG6 activity during NMD[17]. Thus, we hypothesized a direct mechanistic role for this domain in the SMG6 decay mechanism (Fig. 1A). To assess this possibility, we first performed NMD rescue assays in SMG7-deficient Flp-In-T-REx-293 human cells (Fig. 1B). NMD is sensitized in this cellular background as the additional downregulation of SMG5 or SMG6 results in strong NMD inhibition[17,19,20]. We depleted endogenous SMG5 by siRNA-mediated knockdown and re-expressed either full-length SMG5 or a truncated variant lacking the PIN domain (Supplementary Data 1). NMD rescue assays measuring PTC-containing transcript isoforms levels of alternatively spliced SRSF2 or SRSF6 showed that the PIN-deleted SMG5 truncation mutant failed to restore NMD activity in SMG7-SMG5-depleted cells, while the full-length SMG5 construct rescued the NMD defect (Fig. 1C; Supplementary Fig. 1A).

Although SMG5 and SMG6 are not known to form a stable, constitutive complex in cells, we reasoned that a transient interaction in vivo may be recapitulated by computational and in vitro approaches. Accordingly, recent large-scale in silico screens predicted interactions between SMG5 and SMG6, strengthening our hypothesis[21,22]. Computational predictions using AlphaFold Multimer[23,24] or AlphaFold 3[25] consistently positioned the PIN-like folds of SMG6 and SMG5 in an extended side-by-side interaction burying a mean predicted surface area of 677 Å² at the immediate PIN-PIN interface (Fig. 1D, Supplementary Fig. 2A and Supplementary Data 2). In the computational model, the SMG5-SMG6 PIN-PIN dimerization unit appeared to be stabilized by contacts between two α-helices preceding the PIN domains and by an intramolecular interaction between a loop region in the SMG6 TPR-like helical domain and a β-sheet region that flanks the PIN fold in this protein (Fig. 1D, Supplementary Fig. 2A). To assess the potential for a direct interaction, we expressed and purified the putative interacting regions predicted computationally, namely a SMG5 region encompassing the PIN fold and the preceding helix (residues 800-1016) fused to an N-terminal 3xFLAG tag and a C-terminal C-tag (hereafter referred to as FLAG-SMG5$_P$) and a SMG6 construct consisting of the central TPR-like domain and the C-terminal PIN domain (residues 565-1419) fused to an N-terminal TwinStrep tag (hereafter referred to as TS-SMG6$_{TP}$) (Fig. 1E, F). In pull down assays, recombinant FLAG-SMG5$_P$ co-precipitated TS-SMG6$_{TP}$,

albeit in sub-stoichiometric amounts (Fig. 1F), indicating the presence of a direct yet weak interaction. Using crosslinking-mass spectrometry (XL-MS), we found intermolecular crosslinks between the SMG5 PIN domain and the PIN domain as well as the TPR-like domain of SMG6 (Supplementary Fig. 1B, C). When mapping the obtained intermolecular crosslinks onto our AlphaFold 3 model (Supplementary Fig. 1D) none of the three crosslinks meet the distance constraint of 26–30 Å between the Cα atoms of the crosslinked lysine residues. However, we note that the conformation of the SMG6 TPR-like domain with respect to the PIN-PIN interface is divergent among the AlphaFold models, suggesting a conformational flexibility of this domain. Furthermore, the lysine residue (K928) in the SMG5 PIN domain crosslinked to either the PIN domain or the TPR-like domain of SMG6 is located in a flexible, solvent-accessible loop of 18-nt length, which is predicted with low to very low confidence (Supplementary Fig. 3E). The observed intermolecular crosslinks between the SMG5 PIN domain and the two structured domains of SMG6 suggest that the two proteins are in close proximity in solution, thus supporting a direct protein-protein interaction of TS-SMG6$_{TP}$ and FLAG-SMG5$_P$ also in absence of RNA.

We next assessed whether this interaction affects the endonucleolytic activity of the SMG6 PIN domain. To this end, we performed in vitro nuclease assays using a substrate of a single-stranded U$_{30}$ RNA with a fluorescent 6-FAM label at the 5′ end and with the backbone of the first ten nucleotides modified with phosphorothioate linkages (FAM-U*$_{10}$-U$_{20}$). The phosphorothioate modifications are expected to render the backbone largely resistant to nucleolytic degradation by SMG6 or contaminating ribonucleases, thus allowing the accumulation of detectable decay products of approximately ten nucleotides (Fig. 1G). Similar to previous reports[13,14,26,27], TS-SMG6$_{TP}$ alone only exhibited weak nucleolytic activity in vitro, which was even further impaired upon mutation of an active site residue (D1353A; Fig. 1H, lanes 3-5,7; Supplementary Fig. 1E). Owing to the weak nucleolytic activity of SMG6, we used a higher protein-to-RNA ratio that is consistent with conditions routinely applied to other eukaryotic PIN domains in vitro[28,29]. The presence of FLAG-SMG5$_P$ strongly stimulated the nucleolytic activity of wild-type TS-SMG6$_{TP}$ (Fig. 1H, lanes 11-13; Supplementary Fig. 1E) in a specific manner, as it had no effect on the TS-SMG6$_{TP}$ catalytic mutant (Fig. 1H, lane 15; Supplementary Fig. 1E). In nuclease assays performed in a time-course set up, the stimulation of TS-SMG6$_{TP}$ by FLAG-SMG5$_P$ not only accelerated the reaction rate, but also elevated the reactivity to phosphorothioate-modified linkages (Fig. 1H, lanes 11-13; Supplementary Fig. 1E). To confirm that the observed stimulatory effect of FLAG-SMG5$_P$ on TS-SMG6$_{TP}$ was indeed specific to the endonucleolytic activity of SMG6 and not mediated by contaminating nucleases, we repeated the in vitro nuclease assays with a circularized substrate (Supplementary Fig. 1F), which had been generated by self-ligating a single-stranded DNA-RNA hybrid oligonucleotide with an internal Fluorescein label (5′-dT$_7$-iFluorT-dT$_2$-U$_{30}$-3′). As with the linear substrate, we observed weak catalytic activity of the wild-type TS-SMG6$_{TP}$ protein alone, which was strongly stimulated in presence of FLAG-SMG5$_P$ (Supplementary Fig. 1G). Interestingly, the in vitro activity of TS-SMG6$_{TP}$ alone also appeared to be expanded to the deoxyribose-phosphate backbone in presence of FLAG-SMG5$_P$, although the SMG6 PIN domain alone is unable to use single-stranded DNA as a substrate in vitro[13]. We note that the residual activity observed with the catalytic mutant of TS-SMG6$_{TP}$ or with FLAG-SMG5$_P$ alone are most likely background activities originating from contaminating nucleases in the protein preparations.

We concluded that the SMG5 PIN domain is required to enhance the endonuclease activity of the SMG6 PIN domain in cells, and that this effect can be recapitulated in vitro using purified recombinant proteins, which are capable of a direct, albeit weak, interaction under these simplified biochemical conditions.

 

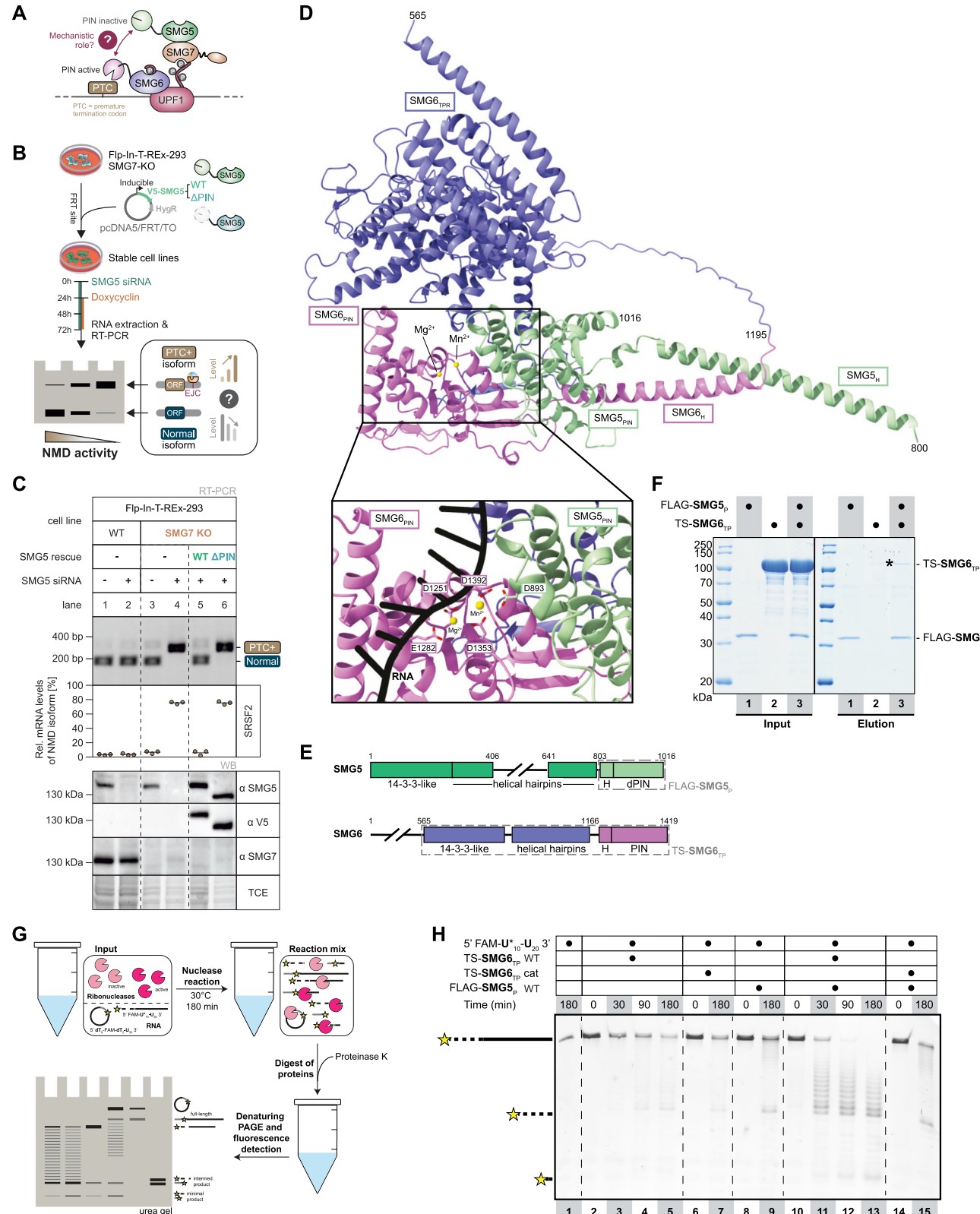

## Composite active site formed by the SMG5 PIN-SMG6 PIN interaction

Computational predictions of SMG6TP-SMG5P in the presence of a single-stranded RNA using AlphaFold 3[25] enabled us to generate precise and testable structural models on the interaction and enzymatic mechanism (Fig. 1D; Supplementary Fig. 3A). The computational models consistently predicted that an 11-nucleotide single-stranded

RNA binds in a defined conformation across the PIN-PIN interface of the SMG5-SMG6 dimer (Fig. 1D; Supplementary Fig. 3A). In the AlphaFold 3 predictions, we also included magnesium and manganese ions to reflect the expected two-metal-ion catalytic mechanism characteristic of PIN domains[30,31]. The resulting three-dimensional model showed that the two metal ions are coordinated by the four canonical acidic residues[12,13,27,32] in the SMG6 PIN domain active site (D1251, E1282,

**Fig. 1 | SMG5 and SMG6 interact via the PIN domains to stimulate endonucleolytic activity. A** Cartoon depicting the final degradation step of NMD. **B** Schematic overview of the NMD rescue assays. SMG7 KO cells were stably transfected with siRNA-resistant SMG5 WT or ΔPIN expression constructs. Cells were treated with SMG5 siRNA to deplete endogenous SMG5 and with doxycycline to induce expression of exogenous SMG5. NMD activity is determined by RT-PCR of alternatively spliced transcripts. **C** Rescue assay of SMG5 WT and SMG5 ΔPIN. Endpoint RT-PCR detection of SRSF2 transcripts was carried out in WT or SMG7 KO cell lines containing SMG5 rescue constructs, combined with SMG5 knockdowns. Upper bands represent NMD-sensitive transcript isoforms whereas lower bands represent canonical transcript isoforms. Relative mRNA levels of the NMD-sensitive isoforms were quantified based on their band intensity ($n = 3$ biologically independent samples). Western blot analysis using anti-SMG5, anti-V5, and anti-SMG7 antibodies revealed successful expression of the SMG5 rescue constructs and validated the knockout of the SMG7 protein. TCE staining served as loading control. Source data are provided as a source data file. **D** Overview of an AlphaFold 3 model of human SMG6$_{TP}$ and SMG5$_P$ forming an extended side-by-side arrangement via their PIN domains and close-up of the active site of the endonuclease. Input: SMG5$_{800-1016}$ + SMG6$_{565-1419}$ + 11-mer RNA (U$_3$-UGAAC-U$_3$) + Mn$^{2+}$ + Mg$^{2+}$. Domains are colored as in A and divalent metal ions are shown as yellow spheres. RNA is omitted for clarity in the overview and shown in black in the close-up. Catalytic residues (SMG6 D1251, E1282, D1353, D1392 and SMG5 D893) are indicated in the close-up. One out of 5 models is displayed. All models converged on the PIN-PIN interaction of SMG5 and SMG6 and the expanded active site. The extended segment of the SMG6 PIN domain is expanded to a three-stranded β-sheet by a long flexible linker in the TPR-like domain of SMG6 (one purple and two pink β-strands).

**E** Schematic representation of domain organization of human SMG5 and SMG6 with folded domains indicated by rectangles and unstructured segments indicated by lines. 14-3-3-like domain and helical-hairpin domain together form a TPR-like domain. The C-terminal PIN domains are both preceded by an α-helix (H). The PIN domain of SMG5 is catalytically dead (d), whereas the PIN domain of SMG6 exhibits endonucleolytic activity. Grey-dashed boxes indicate the parts of the proteins used in in vitro assays, comprising TPR-like and PIN domain with preceding helix of SMG6 (TS-SMG6$_{TP}$) and PIN domain with preceding helix of SMG5 (FLAG-SMG5$_P$), respectively. **F** Coomassie-stained SDS-PAGE analysis of an anti-FLAG pull-down assay with purified constructs of FLAG-SMG5$_P$ and TS-SMG6$_{TP}$. Band indicated by asterisk corresponds to TS-SMG6$_{TP}$ pulled down in sub-stoichiometric amounts by FLAG-SMG5$_P$. The gel shown is representative of two independent experiments, both producing consistent results. **G** Schematic overview of in vitro nuclease assays. A U$_{30}$ RNA, 5′-labeled with 6-FAM fluorophore and modified with phosphorothioate linkages in the first ten nucleotides, or a circularized dT$_{10}$-U$_{30}$ DNA-RNA hybrid with an internal Fluorescein label were used as substrate (5′ FAM-U*$_{10}$-U$_{20}$ 3′ or circ. 5′-dT$_7$-iFluorT-dT$_2$-U$_{30}$-3′). RNA is shown in black with phosphorothioate linkages indicated by dashes, while DNA is represented in grey. Nucleases in large excess were incubated with the substrate. Following digest of nucleases by Proteinase K, the RNA decay intermediates were analyzed via denaturing PAGE and in-gel fluorescence detection. **H** Denaturing PAGE analysis of decay intermediates of in vitro nuclease assay with time-course set up showing catalytic activity of TS-SMG6$_{TP}$ in absence and presence of FLAG-SMG5$_P$. TS-SMG6$_{TP}$ cat represents the catalytically inactive D1353A mutant. Samples were taken at the indicated time points. The gel shown is representative of three independent experiments, each producing the same pattern.

---

D1353, D1392) and, unexpectedly, by an aspartic acid residue in the SMG5 PIN domain (D893), thus suggesting that the SMG5 PIN domain may contribute one of the catalytic residues for effective endonucleolytic activity.

## Conditional depletion reveals essential roles of SMG5 and SMG6 in NMD

Having established that SMG5 directly enhances the catalytic activity of the SMG6 PIN domain in vitro, we next asked whether this dependency is reflected in cellular NMD. Despite long-standing assumptions regarding the essentiality of SMG5 and SMG6, it has remained unclear whether NMD can proceed under near-complete depletion of either factor, in part because siRNA-mediated depletion leaves residual protein sufficient to sustain partial pathway activity[17,19]. Moreover, the essentiality of SMG5 and SMG6 in cells has precluded the generation of conventional knockout cell lines[33]. To circumvent these limitations, we engineered conditional depletion lines in which endogenous SMG5 or SMG6 was fused to FKBP12$^{F36V}$[34], enabling their rapid and inducible degradation using the dTAG$^V$-1 degron system[35] (Fig. 2A). Because both siRNA-mediated and degron-mediated protein depletion typically leave residual protein that limits the extent of NMD inhibition (Supplementary Fig. 4A, lanes 3,4,6,9), we combined dTAG-mediated degradation with siRNA-mediated knockdown, enabling a more complete inhibition of NMD (Fig. 2B; Supplementary Fig. 4A, B). To assess the global impact on NMD under these conditions, we performed mRNA-seq. As a reference for a very efficient NMD inhibition, we also depleted UPF1 using dTAG-engineered cells[36]. Principal component analysis (PCA) showed that SMG5- and UPF1-depleted conditions clustered together, whereas SMG6-depleted samples formed a distinct group (Fig. 2C left). Further analysis of the top 100 genes contributing to the PCA revealed that principal component 1 (PC1) captured largely NMD-relevant effects (Fig. 2C right). Concordantly, loss of SMG5 mirrored UPF1 depletion, with a similar accumulation of NMD targets at the consolidated NMD-regulated human transcriptome (NMDRHT) transcript level[36] (Fig. 2D). SMG6 loss also produced strong NMD inhibition, albeit slightly weaker than SMG5 or UPF1 (Fig. 2D), consistent with its separation in the PCA. Nevertheless, the highest overlap of upregulated NMD-regulated human transcripts was found between all tested conditions, confirming the generally similar trend of NMD

inhibition (Supplementary Fig. 4C). Comparison with published RNA-seq data from SMG7-KO cells with additional SMG5 or SMG6 knockdown[17] revealed comparably strong accumulation of both GENCODE-annotated genes and NMDRHT-annotated transcripts (Supplementary Fig. 4D, E), as well as similar PCA clustering. Visual inspection of representative NMD targets confirmed expected signatures of NMD-inhibited conditions, such as the inclusion of an NMD-activating exon and splicing of an NMD-activating intron in the SRSF2 mRNA (Fig. 2E; Supplementary Fig. 4F). Collectively, these results position SMG5 and SMG6 alongside UPF1 as indispensable components of human NMD, consistent with a model in which both factors act together to promote endonucleolytic decay rather than functioning in parallel pathways.

## SMG6 endonuclease activity in cells requires SMG5-SMG6 interaction

Having established that both SMG5 and SMG6 are essential for NMD in human cells, we next investigated whether this interdependence reflects the requirement for a direct SMG5-SMG6 interaction. To directly test this model, we introduced mutations at the predicted SMG5–SMG6 interface or within the catalytic tetrad of SMG6 (Fig. 3A, B, Supplementary Fig. 5A, B) and assessed the functional consequences of these mutants both in cell-based rescue assays and in vitro activity assays. For the rescue assays, we used the SMG6 degron cell line (cl. 2) (Fig. 2B) and combined this with siRNA-mediated knockdown to minimize residual protein (Fig. 3C). In vitro assays were performed with the linear substrate using the same setup described above (Fig. 1G).

Active-site substitutions in SMG6 (D1353A or D1392A), referred to as SMG6 C1 and SMG6 C2 mutants (Fig. 3A) failed to rescue NMD in cultured cells (Fig. 3D lanes 10-11; Supplementary Fig. 5C, D lanes 10-11). These mutants also abrogated the weak catalytic activity of SMG6 in isolation in line with earlier studies[13,14] as well as precluded the stimulation of the catalytic activity in the presence of SMG5 in vitro (Fig. 3E, lanes 3-4, 10-11). Mutations in SMG6 I1 (L1205A, K1208D, L1212D) target the predicted coiled-coil interface preceding the PIN-PIN domain interface, whereas mutations in SMG6 I2 (V1229E L1230E) aim to disrupt the interface of the PIN-preceding α-helix of SMG6 with the SMG5 PIN. Finally, the SMG6 I3 mutant (V1397R,

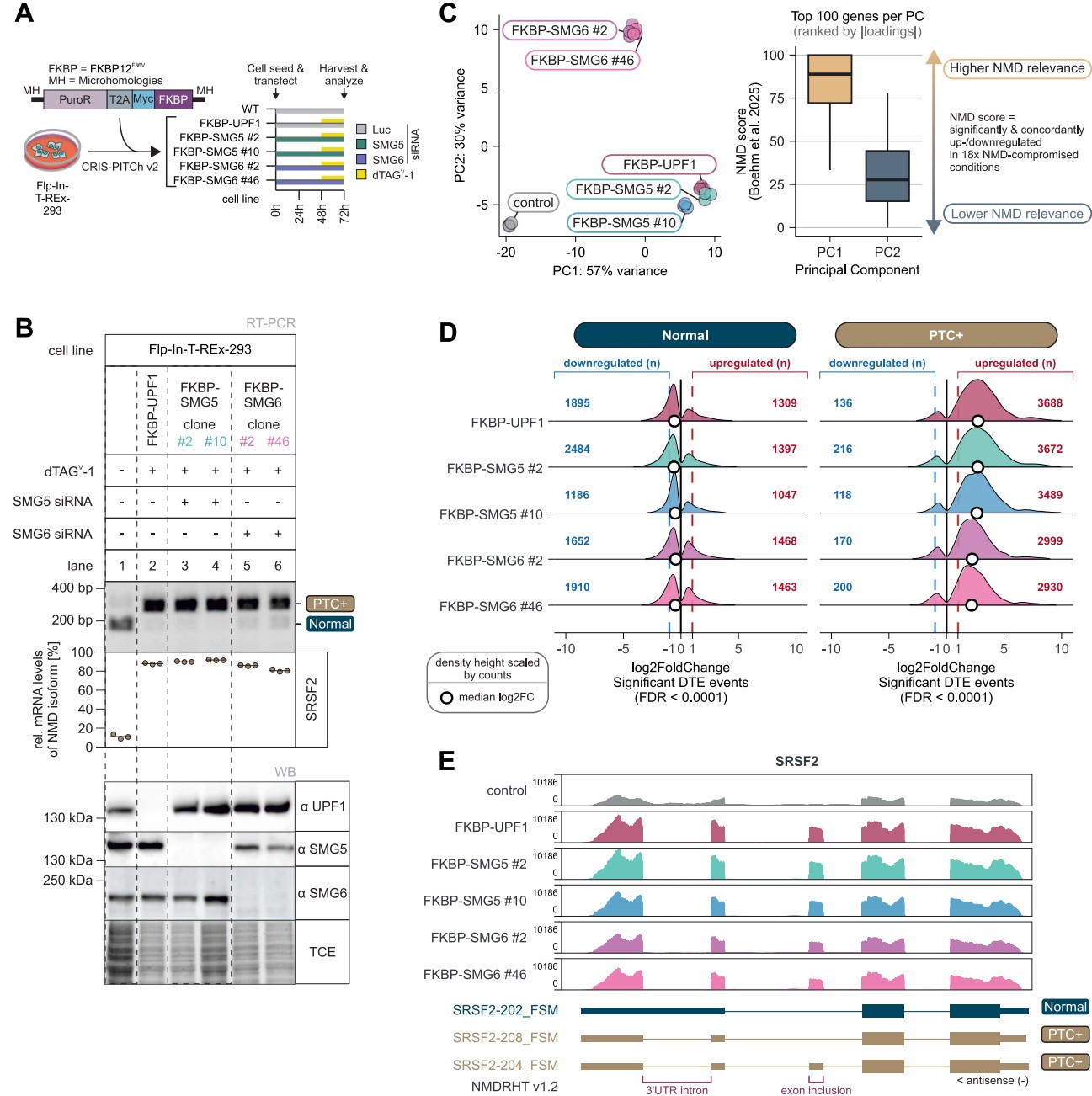

**Fig. 2 | Depletion of SMG5 or SMG6 abolishes NMD in cells. A** Scheme for conditional degron tagging of Flp-In-T-REx-293 cells and experimental setup for protein depletion using combinations of siRNAs and dTAG$^V$-1. **B** NMD activity of degron cells. Endpoint RT-PCR detection of SRSF2 transcript isoforms was carried out in the indicated Flp-In-T-REx-293 degron cell lines, with the addition of dTAG$^V$-1 and the associated knockdowns. Upper bands represent NMD-sensitive transcript isoforms whereas the lower band represents canonical transcript isoforms. Relative mRNA levels of the NMD-sensitive isoforms were quantified based on their band intensity ($n = 3$ biologically independent samples). Western blot analysis using anti-UPF1, anti-SMG5 and anti-SMG6 antibodies revealed successful degradation of the respective protein. TCE staining served as loading control. Source data are provided as a source data file. **C** Gene-level principal component analysis (PCA) of transcriptome profiles from RNA-seq data ($n = 3$ biologically independent samples) of control cell lines and cells with UPF1, SMG5 or SMG6 depletion (left). NMD relevance scores (from Boehm et al. 2025) for the top 100 genes contributing to

principal components 1 and 2 (right). The centerline of the boxplot represents the 50th percentile (median), whereas the lower and upper box limits correspond to the 25th and 75th percentiles. The whiskers extend from the box limits to the smallest or largest value no further than 1.5 * inter-quartile range. Data beyond the end of the whiskers (outliers) are not shown. **D** Expression changes of normal (non-NMD; left) or PTC-containing (right) NMDRHT-annotated transcripts in the FKBP-UPF1/SMG5/SMG6 degron cell lines. Only significant Differential Transcript Expression (DTE) events are shown (cutoff FDR < 0.0001). The absolute number of up- or downregulated (|log2FC| >1) transcripts is indicated, the median log2FC is shown as white points and the density height of the ridge plot is scaled by number of transcripts. **E** Read coverage of SRSF2 from RNA-seq data of control and UPF1/SMG5/SMG6-FKBP degron cell lines are shown as Integrative Genomics Viewer (IGV) snapshots. The NMDRHT-annotated isoforms and their predicted NMD status is schematically indicated below.

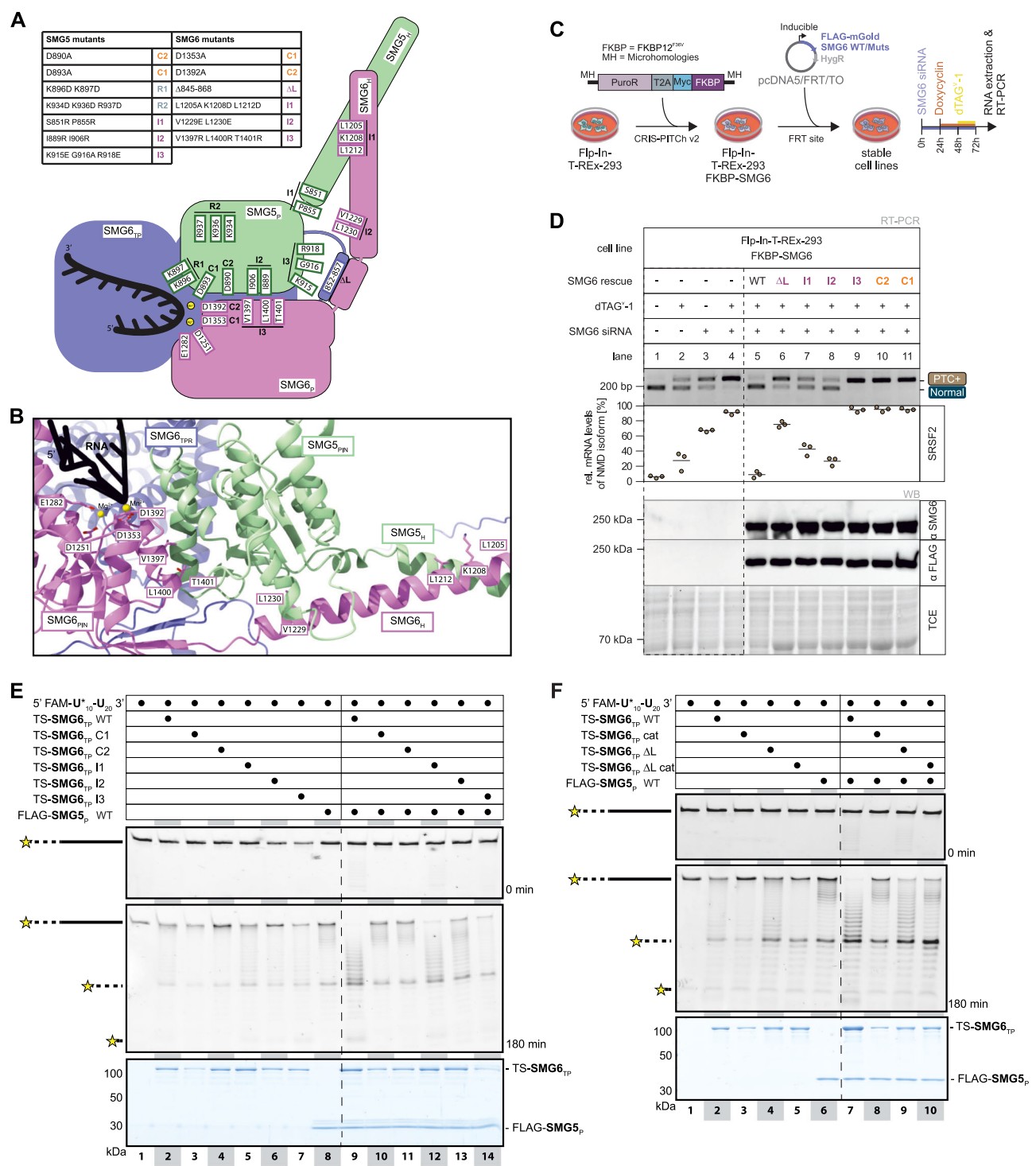

L1400R, T1401R) contains mutations in a central helix at the PIN-PIN interface (Fig. 3A; Supplementary Fig. 5A, B). All three interaction mutants of SMG6 (SMG6 I1-I3) showed defects in NMD rescue activity when expressed in SMG6 degron cells, with I3 being most impaired, I1 moderately affected, and I2 retaining the most activity (Fig. 3D, lanes 7-9; Supplementary Fig. 5C, D, lanes 7-9). In nuclease assays in vitro, the SMG6 I3 mutation impaired the stimulation of SMG6 catalytic activity by SMG5 (Fig. 3E, lane 14), while it had no effect on the activity of SMG6 alone (Fig. 3E, lane 7). Qualitatively, the I1 and I2 mutants had weaker yet reproducible effects on the stimulated catalytic activity in vitro (Fig. 3E, lanes 12-13), consistent with their cellular NMD phenotypes.

Closer inspection of the computational models (Fig. 1E, F; Supplementary Fig. 2A, 3A) showed that the intramolecular interaction within SMG6 involves the extended segment of the SMG6 PIN domain (res. 1239-1244 and 1365-1384)[27] as well as part of the flexible linker connecting the 14-3-3-like domain with the helical hairpins domain in the TPR-like region. This linker is unusually long compared to structurally related proteins, such as SMG7[4] and predicted to be disordered in isolation. Although both elements are highly conserved across metazoan homologs (Supplementary Fig. 1H, I), they are unique features of the SMG6 endonuclease[4,13,27]. We assessed the impact of the intramolecular interaction between the loop region in the TPR-like helical domain and the extended β-sheet region in the PIN domain of

**Fig. 3 | SMG6 activity is stimulated by SMG5. A** Structural framework underlying the design of SMG5 and SMG6 mutants based on AlphaFold 3 models. Relative positions of the PIN domains and linker regions of SMG5 and SMG6 are depicted. All mutations and catalytic residues (SMG6 D1251, E1282, D1353, D1392) are shown at their corresponding structural elements and grouped by functional class. Mutants putatively impairing catalysis (Cx; orange), RNA binding (Rx; blue-grey) or protein-protein interaction (Ix; purple) are summarized in the table (top left). **B** Close-up on extended active site and PIN-PIN interface formed by SMG5-SMG6 as predicted by AlphaFold 3 and displayed in Fig. 1D, with domains colored as in Fig. 1, metal ions shown as yellow spheres, and RNA depicted in black. Catalytic (D1251, E1282, D1353, D1392) and mutated residues of SMG6 are displayed as sticks. **C** Schematic overview of NMD rescue assays. SMG6 degron cells were generated using the CRIS-PITCh method, enabling genomic insertion of the FKBP12$^{F36V}$ degradation tag into the endogenous SMG6 locus. Newly generated SMG6 degron cells were stably transfected with siRNA-resistant SMG6 WT or mutant expression constructs and subsequently treated with SMG6 siRNA and dTAG$^V$-1 to deplete endogenous SMG6 and with doxycycline to induce expression of exogenous SMG6. NMD activity is determined by RT-PCR of alternatively spliced transcripts. **D** Rescue assay of SMG6 mutants in SMG6 degron cells. Endpoint RT-PCR detection of SRSF2 transcripts was carried out in FKBP-SMG6 cells containing SMG6 rescue constructs, combined with SMG6 knockdowns and dTAG$^V$-1 treatment. Upper bands represent NMD-sensitive transcript isoforms whereas lower bands represent canonical transcript isoforms. Indicated fragment sizes (bp) are estimated. Relative mRNA levels of the NMD-sensitive isoforms were quantified based on their band intensity ($n$ = 3 biologically independent samples). Western blot analysis using anti-SMG6 or anti-FLAG antibodies revealed successful expression of SMG6 rescue constructs. TCE staining served as loading control. Source data are provided as a source data file. **E, F** Denaturing PAGE analysis of in vitro nuclease assay with mutants of TS-SMG6$_{TP}$ (**E**) or with Δloop constructs of TS-SMG6$_{TP}$ (**F**) to assess their effect on endonucleolytic activity. Samples taken after 0 min (top) and 180 min (middle) show RNA input and the pattern of decay intermediates, respectively. Coomassie-stained SDS-PAGE analysis of proteins used in assays served as loading control (bottom). The gels shown are representative of three independent experiments, each producing the same pattern.

SMG6 using a loop deletion construct (SMG6 ΔL; deletion of residues 845-868). Although the effect on the catalytic activity in vitro was not pronounced (Fig. 3F), the loop deletion substantially impaired NMD in rescue assays in vivo (Fig. 3D, lane 6; Supplementary Fig. 5C, D, lane 6). Thus, the loop is not required for catalytic activity per se but is important for SMG6 function in cells, in line with a proposed function of the extended segment of the SMG6 PIN domain in correctly positioning the PIN domain with respect to other domains of SMG6[12,27].

### SMG5 completes the composite NMD endonuclease active site

Next, based on our model, we evaluated the contribution of SMG5 to SMG6 endonucleolytic activity on the molecular level through mutational studies (Fig. 4A; Supplementary Fig. 6A). A SMG5 D893A substitution (referred to as SMG5 C1), targeting the residue expected to contribute towards catalytic activity failed to support the stimulation of the endonucleolytic activity in nuclease assays (Fig. 4B, lane 10). Consistently, this mutant was unable to restore NMD in vivo in rescue assays performed in SMG5 degron cell lines (cl. 10), in which endogenous SMG5 was depleted prior to re-expression of the mutant protein (Fig. 4C, lane 7; Supplementary Fig. 6B, C, lane 7). In contrast, a nearby substitution, SMG5 D890A (SMG5 C2), which alters an acidic residue adjacent to but not within the predicted catalytic site, fully rescued NMD function in cells (Fig. 4C, lane 6; Supplementary Fig. 6B, C, lane 6). Together, the cell-based and in vitro data thus support the model that an acidic residue in SMG5 complements the SMG6 active site, thereby enabling the full activation of the endonuclease.

Analysis of the computational model indicated the presence of a highly conserved, positively-charged surface patch adjacent to the active site (Supplementary Fig. 3B). In particular, SMG5 K896 and K897 were predicted to contact the sugar-phosphate backbone of the RNA (Fig. 4A; Supplementary Fig. 6A), suggesting they may be involved in correctly positioning the RNA substrate for endonucleolytic cleavage. We engineered reverse-charge mutations of the putative RNA-binding patch on SMG5 (K896D and K897D, referred to as SMG5 R1). In nuclease assays, SMG5 R1 largely abolished the stimulation of endonucleolytic activity in vitro (Fig. 4B, lane 11), consistent with a defect in RNA substrate binding. In agreement with these biochemical data, SMG5 R1 also reduced NMD efficiency in rescue assays (Fig. 4C, lane 8; Supplementary Fig. 6B, C, lane 8). Together, the cell-based and in vitro results support a model in which conserved positively-charged residues in SMG5 help position the RNA substrate in the active site to facilitate endonucleolytic cleavage.

To validate the predicted protein-protein interaction, we introduced mutations designed to disrupt the SMG5-SMG6 interfaces observed in the computational model and tested their functional implication in vivo and in vitro (Fig. 4A; Supplementary Fig. 6A). The SMG5 I2 mutant (I889R, I906R) putatively disrupts hydrophobic contacts with the central helix of SMG6 at the PIN-PIN interface and is therefore targeting the complementary interface to SMG6 I3. SMG5 I3 (K915E, G916A, R918E) contains mutations in a linker in the SMG5 PIN domain predicted to bind in a negatively charged pocket formed by the TPR, the α-helix preceding the PIN domain and the extended structure of the PIN domain of SMG6. In vitro, FLAG-SMG5$_P$ I2 strongly impaired the stimulation of the catalytic activity of SMG6 by SMG5 in nuclease assays, while FLAG-SMG5$_P$ I3 had a milder effect (Fig. 4B, lanes 12-13). In cell-based rescue assays in the degron cell line, SMG5 I2 was unable to rescue NMD, while SMG5 I3 exhibited a partial rescue with some residual activity (Fig. 4C, lanes 9–10; Supplementary Fig. 6B, C, lanes 9-10), consistent with their behavior in vitro. Overall, we conclude that impairing the PIN-PIN interface (SMG5 I2 and SMG6 I3) exerts a strong effect on both NMD in cells and on the stimulation of the SMG6 catalytic activity by SMG5 in vitro, albeit to differing extents for each mutant. These results support a model that the SMG5-SMG6 interaction is driven largely by contacts between the two PIN domains.

Whereas the cell-based assays described above were performed in SMG5 degron cell lines, we next tested all SMG5 mutants in rescue assays using SMG7-knockout cells. This represents a sensitized background and provides a complementary and more stringent assay for SMG5 function, owing to the ability of SMG7 to attenuate the loss of SMG5[17]. In this system, we observed the same qualitative trends as in the SMG5 degron cell lines, although the defects were generally more pronounced, with NMD being more strongly inhibited (Fig. 4D, Supplementary Fig. 6D, E). However, SMG5 C2 efficiently rescued the NMD defect to wild-type levels in SMG7-depleted cells (Fig. 4D, lane 6; Supplementary Fig. 6D, E, lane 6), consistent with the structural interpretation. To further assess the RNA-binding interface, we introduced substitutions in a neighboring cluster of positively charged residues (K934D, K936D, R937D; SMG5 R2) that lies adjacent to the active site but, unlike the R1 patch, is not predicted to contact RNA. SMG5 R2 completely rescued NMD, even in the sensitized SMG7-knockout background (Fig. 4D, lane 9; Supplementary Fig. 6D, E, lane 9). A mutation targeting a secondary protein-protein interface (SMG5 I1; S851R, P855R), which disrupts contacts between the SMG5 linker and the α-helix preceding the SMG6 PIN domain, caused only a modest effect on NMD (Fig. 4D, lane 10; Supplementary Fig. 6D, E, lane 10). Thus, across both cellular systems and biochemical assays, our findings demonstrate that SMG5 promotes NMD by engaging the SMG6 PIN domain and complementing its active site, thereby enabling full activation of the endonuclease.

## Discussion

SMG6 is an active ribonuclease performing the endonucleolytic cleavage of NMD substrates using its PIN domain, thereby targeting them for exonucleolytic degradation by general decay machineries[13–15]. Although it had previously been reported that the PIN domain of SMG5

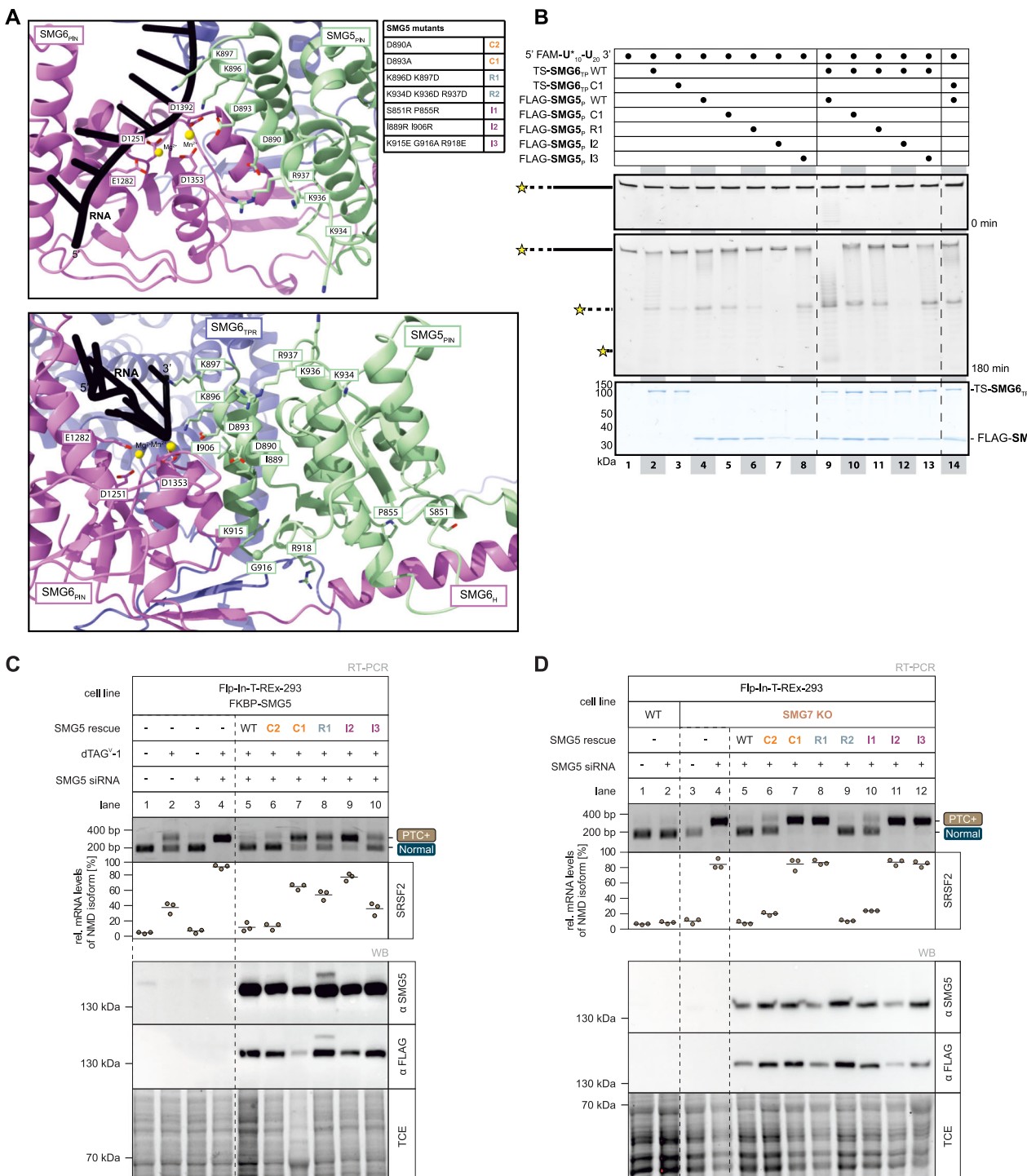

**Fig. 4 | Mutational analysis reveals key residues of SMG5 required for cPIN formation and catalytic activity. A** Close-up on extended active site and PIN-PIN interface formed by SMG5-SMG6 as predicted by AlphaFold 3 and displayed in Fig. 1D, with domains colored as in Fig. 1, divalent metal ions shown as yellow spheres, and RNA depicted in black. Two different orientations are represented. Catalytic residues (D1251, E1282, D1353, D1392) of SMG6 and mutated residues of SMG5 are displayed as sticks. SMG5 mutants putatively impairing catalysis (Cx; orange) or RNA binding (Rx; blue-grey) or disrupting protein-protein interaction (Ix; purple) are summarized in the table (right). **B** Denaturing PAGE analysis of in vitro nuclease assay with mutants of FLAG-SMG5$_P$ to assess their effect on endonucleolytic activity. Samples taken after 0 min (top) and 180 min (middle) show RNA input and the pattern of decay intermediates, respectively. Coomassie-stained SDS-PAGE analysis of proteins used in assay served as loading control

(bottom). The gels shown are representative of three independent experiments, each producing the same pattern. Source data are provided as a source data file. **C**, **D** Rescue assay of SMG5 mutants in SMG5 degron or SMG7 KO cells. Endpoint RT-PCR detection of SRSF2 transcript isoforms was carried out in Flp-In-T-REx-293 SMG5 degron cells (**C**) or SMG7 KO cells (**D**) containing SMG5 rescue constructs, combined with dTAG$^V$-1 treatment and/or SMG5 knockdowns. Upper bands represent NMD-sensitive transcript isoforms whereas lower bands represent canonical transcript isoforms. Indicated fragment sizes are estimated. Relative mRNA levels of the NMD-sensitive isoforms were quantified based on their band intensity ($n = 3$ biologically independent samples). Western blot analysis using anti-SMG5 or anti-FLAG antibodies revealed successful expression of the SMG5 rescue constructs. TCE staining served as loading control.

**A**

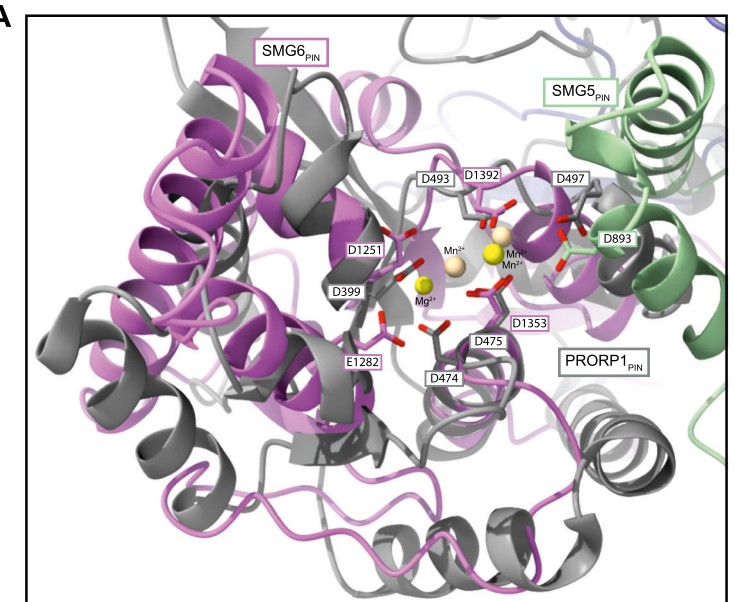

**B**

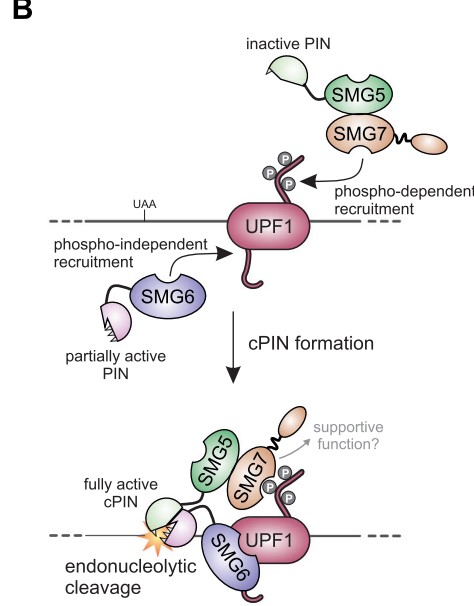

**Fig. 5 | Comparison of SMG5-SMG6 cPIN with PRORP1 and model for its role in NMD. A** Overlay of the computational model of the composite SMG5-SMG6 PIN domain predicted by AlphaFold 3 with a crystal structure of the PIN domain of Proteinaceous RNase P1 (PRORP1) from *A. thaliana* (PDB: 4G24; residues: 358-533)[61]. Catalytic residues of the PIN domains are displayed as sticks and labeled (SMG5-SMG6 cPIN: SMG6 D1251, E1282, D1353, D1392 and SMG5 D893; PRORP1: D399, D474, D475, D493, D497). Divalent metal ions are shown as spheres. SMG5 and SMG6 are colored as in Fig. 1D, while PRORP1 is colored in grey. RNA in computational model is omitted for clarity. **B** Model for the formation of the composite SMG5-SMG6 PIN domain as the NMD endonuclease. After recognition of the NMD substrate by UPF1, the downstream NMD effectors are recruited for the ultimate degradation of the substrate. While SMG6 binds UPF1 in a phosphorylation-independent manner, SMG5-SMG7 are recruited in a phosphorylation-dependent manner, enabling the formation of the composite PIN (cPIN) from the partially active PIN domain of SMG6 and the inactive PIN domain of SMG5. The fully activated composite enzyme carries out the endonucleolytic cleavage of the NMD target initiating the ultimate degradation of the transcript.

is required for cleavage of NMD substrates by SMG6 in the cellular context[17], the exact function of the catalytically inactive PIN domain of SMG5 has remained elusive.

Using a combination of structure prediction, biochemical and cell-based assays, we show that the SMG5 PIN domain directly interacts with the SMG6 PIN domain, thereby considerably stimulating the catalytic activity of the latter. The stimulatory effect of SMG5 is likely brought about by an expansion of the SMG6 active site by an aspartate residue in SMG5 (D893) as well as by the contribution of the SMG5 PIN domain to the positioning of the RNA substrate in the SMG6 active site (Fig. 5A). Mutations of these functionally important residues as well as of critical residues at the PIN-PIN interface not only abolish the stimulatory effect of SMG5 on the SMG6 catalytic activity in vitro, but also impair NMD in rescue assays in cells. These results rationalize the weak nucleolytic activity of SMG6 in isolation in vitro[13,14] as well as the requirement of the SMG5 PIN domain, which is itself catalytically inactive, for SMG6-mediated endonucleolytic cleavage in vivo[17] and the heterodimer-independent function of SMG5 in cells[37,38]. As a note, the catalytic residue provided by SMG5 (D893) could indeed be the functional equivalent of a fifth catalytic residue present in some active PIN domains, such as *Arabidopsis thaliana* Proteinaceous RNase P-1 (PRORP1) (Fig. 5A) or *Mycobacterium tuberculosis* VapC15[12,32].

Taken together, our data support a model in which the endonucleolytic cleavage events on NMD substrates are tightly controlled by a reconciliation of the phosphorylation-independent recruitment of the SMG6 endonuclease[4,26,39] and the phosphorylation-dependent recruitment of the SMG5-SMG7 heterodimer[4,5,10,26] via the direct interaction of the PIN domains of SMG6 and SMG5 (Fig. 5B). This interaction is expected to be transient and to precisely unleash a potent endonuclease activity at the correct time and place in the NMD pathway. The mechanism also rationalizes how cells maintain the NMD endonuclease activity at bay when not needed, with an incomplete active site when SMG6 and SMG5 are not brought into proximity via their interactions with UPF1.

Although our results establish how SMG5 contributes to SMG6 activation, the role of SMG7 in this process remains unresolved. SMG7 does not seem to contribute catalytic activity, yet its partial redundancy with SMG5 in cells suggests that it may exert a supportive function, potentially stabilizing the UPF1-SMG5-SMG6 complex, modulating its phosphorylation-dependent assembly, or fine-tuning the timing of endonucleolytic cleavage. Defining the molecular contribution of SMG7 will require future structural and functional studies. Beyond the specific case of SMG5 and SMG6, the composite PIN module they form exemplifies how paralogous nuclease folds can cooperate to optimize catalysis. Such division of labor, where one PIN domain contributes catalytic chemistry and RNA positioning while the other provides the primary active site, may represent a recurrent strategy in RNA surveillance machines, considering that both active and inactive PIN domains are found in various complexes in RNA processing, turnover and quality control (such as in Nmd4, Nob1 and Rrp44/DIS3). It also offers a conceptual framework for understanding how transient assemblies can produce switch-like enzymatic outputs in cellular pathways.

## Methods
### Cloning
Point mutations in SMG5 and SMG6 were introduced by site-directed mutagenesis PCR using Q5 high fidelity DNA polymerase (NEB), following the manufacturer's protocol (primers listed in Supplementary Data 1). The mutated fragments were cloned into the tetracycline-inducible pcDNA5/FRT/TO vector via double digestion with NheI and NotI restriction enzymes (ThermoScientific) and subsequent ligation. For protein detection, SMG5 constructs were fused with an N-terminal V5- or FLAG-tag, while SMG6 constructs carried an N-terminal FLAG-mGold-tag.

For generation of proteins used in in vitro experiments, the respective parts of the open reading frames of human SMG5 (UniProt: Q9UPR3; residues 800–1016) or human SMG6 (UniProt: Q86US8 isoform 1; residues 565–1419) with the wild-type sequence or the indicated mutations were cloned into a pBR322-derived vector. SMG5 constructs were designed with an N-terminal 3xFLAG tag followed by a 3C protease cleavage site and a C-terminal C-tag. SMG6 constructs carried an N-terminal TwinStrep tag followed by a 3C protease cleavage site. Point mutations were introduced by site-directed mutagenesis PCR and verified by DNA sequencing.

## Protein expression and purification

TwinStrep-3C-tagged SMG6 fragments were recombinantly expressed in *E. coli* Star (DE3) pRARE. Cultures were grown to an optical density at 600 nm ($OD_{600}$) of 1.0-1.5 at 37 °C and 180 rpm, before the temperature was shifted to 18 °C and protein expression was induced by addition of 0.5 mM IPTG. After incubation over-night, cells were harvested by centrifugation at 8,983 x g and resuspended in Buffer A (50 mM HEPES-KOH, pH 7.5; 500 mM NaCl; 2 mM DTT) supplemented with 1 mM AEBSF (PanReac AppliChem), DNase I (Roche), Benzonase (Merck), 2 mM $MgCl_2$, BioLock (IBA Lifesciences), and 0.1 mg/mL lysozyme. The cell suspension was lysed by ultrasonication and the cell debris was pelleted by centrifugation at 75,600 x g and 4 °C for 30 min. The cleared lysate was passed over a 5 mL StrepTrap XT column (Cytiva) pre-equilibrated with Buffer A for affinity chromatography. The column was washed with Buffer A and Buffer B (20 mM HEPES-KOH, pH 7.5; 50 mM KCl; 1000 mM NaCl; 10 mM $MgSO_4$; 2 mM ATP; 10% (*v/v*) glycerol) and again Buffer A, before bound proteins were eluted with Buffer A supplemented with 50 mM biotin. The wash step with ATP and a high salt concentration (Buffer B) was included with the goal to remove co-purifying chaperones as well as nucleic acids and contaminating RNases. The eluate was concentrated using Amicon Ultra-15 centrifugal filters (Merck) with a molecular weight cut-off of 30 kDa and subjected to size-exclusion chromatography over a HiLoad 16/600 Superdex 200 pg column (Cytiva) in gel filtration buffer (20 mM HEPES-KOH, pH 7.5; 150 mM NaCl; 1 mM DTT). Peak fractions containing the protein of interest were pooled and concentrated using Amicon Ultra-4 centrifugal filters (Merck) with a molecular weight cut-off of 30 kDa. The concentrated protein was flash-frozen in liquid nitrogen and stored at −80 °C until use. To reduce the possibility of introducing RNases into the protein preparations, the purification of constructs used in in vitro nuclease assays was carried out using RNase-free buffers and reagents as well as sterilized reaction tubes and the gel filtration columns and the connected injection loops were cleaned with sodium hydroxide prior to injection of the concentrated eluate from the affinity step.

SMG5 fragments tagged with an N-terminal 3xFLAG-3C tag and a C-terminal C-tag were recombinantly expressed and harvested as described above. Cells were resuspended in Buffer A (50 mM HEPES-KOH, pH 7.5; 500 mM NaCl; 2 mM DTT) supplemented with 1 mM AEBSF, DNase I, Benzonase, 2 mM $MgCl_2$, and 0.1 mg/mL lysozyme and lysed by ultrasonication. The cell debris was pelleted by centrifugation at 75,600 x g and 4 °C for 30 min. The cleared lysate was passed over a 1-mL CaptureSelect XL (ThermoScientific) column pre-equilibrated with Buffer A for affinity chromatography. The column was washed with Buffer A and Buffer B (20 mM HEPES-KOH, pH 7.5; 50 mM KCl; 1000 mM NaCl; 10 mM $MgSO_4$; 2 mM ATP; 10% (*v/v*) glycerol) and again Buffer A, before bound proteins were eluted with Buffer C (20 mM Tris-HCl, pH 7.5; 2 M $MgCl_2$). The eluate was dialyzed against gel filtration buffer (20 mM HEPES-KOH, pH 7.5; 150 mM NaCl; 1 mM DTT) over-night and subsequently concentrated using Amicon Ultra-4 centrifugal filters (Merck) with a molecular weight cut-off of 10 kDa. For size-exclusion chromatography, the concentrated protein preparation was run over a Superdex 200 increase 10/300 GL column (Cytiva) in gel filtration buffer. Peak fractions containing the protein of

interest were pooled and concentrated using Amicon Ultra-4 centrifugal filters (Merck) with a molecular weight cut-off of 10 kDa. The concentrated protein was flash-frozen in liquid nitrogen and stored at −80 °C until use. For SMG5 constructs used in in vitro nuclease assays, similar precautions against introducing RNase contaminations were taken as described for the SMG6 constructs.

## In vitro pull-down experiments

In vitro pull-down experiments with SMG5 constructs were performed with magnetic Protein G DynaBeads (Invitrogen) pre-bound to anti-FLAG M2 antibody (Sigma-Aldrich).

2 μM of 3xFLAG-tagged bait protein (FLAG-SMG5$_P$) and a five-fold molar excess of TwinStrep-tagged prey protein (TS-SMG6$_{TP}$) were mixed in a total volume of 40 μL with binding buffer (50 mM potassium phosphate, pH 6.4; 80 mM NaCl; 2 mM $MgCl_2$; 1 mM DTT; 0.1% NP-40). Protein mixtures were incubated for 60 min at 12 °C under continuous shaking to allow for complex formation, before input samples (4 μL) were set aside for analysis by SDS-PAGE. 20 μL of pre-equilibrated anti-FLAG-bound DynaBeads were added to the mixtures, which were diluted to 100 μL with binding buffer and subsequently incubated for at least 30 min at 4 °C on a rotating wheel. Following the removal of the flow-through, the beads were washed four times with 200 μL binding buffer. For elution, the washed beads were resuspended in 15 μL elution buffer (0.3 mg/mL 3xFLAG peptide in binding buffer) and incubated for 25 min at room temperature. Input samples and eluates were mixed with SDS sample buffer, boiled for 5 min at 95 °C and analyzed by SDS-PAGE.

## In vitro nuclease assay

Fluorescently labeled RNA substrates (Integrated DNA Technologies) for in vitro nuclease assays were gel-purified after migration on a 22% acrylamide (19:1)/ 5 M urea gel in 1x TBE. The purified fluorescently labeled oligonucleotides were dissolved in RNase-free water and stored at −20 °C until further use.

The in vitro degradation assays with SMG5 and SMG6 were performed in reaction buffer (20 mM HEPES-KOH, pH 7.5; 50 mM KCl; 10 mM $MnCl_2$; 2 mM DTT) for 180 min at 30 °C and 350 rpm in the dark. 20 nM of fluorescently labeled substrate was mixed with 3.6 μM of each protein (180x excess of protein over substrate). For analysis of the degradation pattern, 4 μL of the reaction mix was taken at the indicated time points and immediately mixed with an equal volume of stop buffer (50 mM EDTA; 0.1% SDS), before 0.4 U Proteinase K (New England Biolabs) was added and the sample was incubated for 15 min at 37 °C to digest the proteins. Following treatment with Proteinase K, the samples were diluted with 11.5 μL loading dye (10 mM EDTA and 0.1% (*w/v*) Orange G in formamide) and denatured for 5 min at 70 °C. 6 μL of the denatured samples were loaded onto a 22% acrylamide (19:1)/ 5 M urea gel in 1x TBE and denaturing PAGE was run at 10 W for 10–15 min. Gels were imaged using a Typhoon FLA 7000 imager (GE Healthcare) for detection of FAM fluorescence (excitation: 495 nm; emission: 517 nm). For protein loading controls, 4 μL of the reaction mix was taken after 180 min, mixed with SDS sample buffer, boiled for 5 min at 95 °C and analyzed by SDS-PAGE. Circular substrates were obtained by self-ligating a linear DNA-RNA hybrid containing an internal Fluorescein label for fluorescent detection. To this end, 7.5 μM of the linear nucleic acid and 12.5 μM ATP were incubated with 0.5 U/μL T4 polynucleotide kinase (New England Biolabs) and 0.5 U/μL T4 RNA Ligase 1 (New England Biolabs) in the manufacturer-provided reaction buffer supplemented with 10% PEG-8000 at 16 °C over-night. After digest of the enzymes with Proteinase K, the circularized products were gel-purified as described above. To test the successful circularization, 0.63 μg of the nucleic acid substrate were digested with 0.63 U of Ambion™ RNase I (Invitrogen) or RNase R (New England Biolabs) in RNase R reaction buffer for 30 min at 37 °C. Following treatment with

Proteinase K, the reaction mixes were analyzed via denaturing PAGE as described above. Fluorescently labeled RNA substrates and commercially available enzymes used with the RNA substrates are summarized in Supplementary Data 1.

## Cell culture

Human embryonic kidney (HEK) Flp-In-T-REx-293 cells (Thermo Fisher Scientific, cat. no. R78007; RRID: CVCL_U427) were cultivated in high glucose, GlutaMAX DMEM (Gibco) supplemented with 10% fetal bovine serum (Gibco) and 1X Penicillin-Streptomycin (Gibco). The cells were cultured at 37 °C and 5% $CO_2$ in a humidified incubator. The generation of knock-in/knock-out and stable cell lines is described below. All cell lines are summarized in Supplementary Data 1.

## Generation of SMG7 KO cell lines using CRISPaint

SMG7 knockout (KO) Flp-In-T-REx-293 cells were generated using the CRISPR-assisted insertion tagging system (CRISPaint)[40], where a stop cassette was inserted at the Cas9-induced double-strand break to disrupt gene function. The sgRNA sequence for SMG7 KO was 5′-CACCGTCTAAAGCGTATTCCAAAT-3′. $2.8 \times 10^5$ cells per sgRNA were seeded in 6-well plates and transfected the following day. Transfection was performed using 1.2 μg universal donor plasmid (pCRISPaint-myc-HygR), 1.2 μg frame/target selector (pX330-SMG7-R3-mCherry_Cas9-Frame+2) and Lipofectamine 2000 (Thermo Fisher Scientific) according to the manufacturer's instructions. The frame/target selector plasmid carries a stop cassette with a frame selector to ensure correct in-frame integration at the cleavage site. 48 h post-transfection, cells were trypsinized and replated in 10 cm dishes, followed by selection with 1 μg/ml puromycin (InvivoGen). Individual cell colonies were expanded and screened for genome editing by PCR and Sanger sequencing (Eurofins Genomics). SMG7 protein expression was analyzed by western blotting analysis.

## Generation of degron cell lines using CRIS-PITCh v2

For the tagging of endogenous SMG5 or SMG6 with FKBP12$^{F36V}$ (FKBP), a knock-in using the CRIS-PITCh v2 system was performed[41]. Flp-In-T-REx-293 WT or SMG7-depleted cells were seeded at a density of $2.8 \times 10^5$ cells per well of a 6-well plate. One day after seeding the medium was exchanged and the following plasmids were transfected using Lipofectamine 2000 (Thermo Fisher Scientific): 1.2 μg pX330-PITCh-SMG5/SMG6-gRNA encoding the gene specific sgRNA (SMG5: 5′-CACCGAAGCGGCTTTACCGGTGAGA-3′; SMG6: 5′-CACCGCT-CAACCGATTCCTTAGACG-3′) and 0.6 μg pCRIS-PITChv2-PurR-FKBP (SMG5/SMG6 Nter) donor plasmid containing the desired insert. The pCRIS-PITChv2 vector harbors two N-terminal SMG5/SMG6 microhomologies that flank a puromycin resistance gene, a T2A signal, a myc-tag, the FKBP12 tag, and a linker region. The medium was exchanged the next day. Three days after transfection, the medium was supplemented with 1 μg/ml puromycin (InvivoGen) to select for successful knock-ins. Surviving colonies were screened for correct integration via genotyping PCR, followed by Sanger sequencing (Eurofins Genomics).

## Stable cell line generation using Flp-In T-Rex Transfection

24 h before transfection, $2.8 \times 10^5$ cells were seeded per 6-well. One hour before transfection, medium was changed. The cells were then stably transfected with 1–2 μg pcDNA5/FRT/TO expression vector containing mutated SMG5/SMG6 constructs together with 1 μg Flp recombinase expressing plasmid pOG44 via the calcium phosphate-based method with BES-buffered saline (BBS). All used plasmids are summarized in Supplementary Data 1. 48 h after transfection, cells were transferred into 10 cm dishes and selection of cells were conducted using 100 μg/ml hygromycin (InvivoGen). After two weeks, colonies were pooled. To induce the expression of the stably integrated constructs, 1 μg/ml doxycycline was added.

## siRNA-mediated knockdowns (rescue experiments)

Cells were seeded in 6-well plates at a density of $2.8 \times 10^5$ cells per well. Reverse transfection was performed using 2.5 μl Lipofectamine RNAi-MAX (Invitrogen) combined with 60 pmol of the respective siRNA according to manufacturer's instructions. All siRNAs used in this study are listed in Supplementary Data 1. Cells were harvested 72 h after reverse transfection.

## Genomic DNA extraction for genotyping

One day prior to extraction of genomic DNA, cells were seeded into a 48-well plate. The extraction was performed with 50 μl QuickExtract DNA Extraction Solution (Lucigen) according to manufacturer's instructions. Clones were genotyped for heterozygous or homozygous cassette knock-in by PCR using MyTaq Red Mix (Bioline) and primer sets specific for the wild-type (WT) or knock-in (KI) alleles.

## RNA extraction

Cells were harvested in 1 ml in-house prepared TRI-reagent per well[42]. RNA was then extracted according to standard TRI-reagent protocols. Phase separation was initiated by 150 μl 1-Bromo-3 Chloropropane (Sigma-Aldrich) and the final RNA pellet was dissolved in 20 μl RNase-free water by 10 min incubation on a shaking 65 °C heat block. Final RNA concentration was measured using an Implen NanoPhotometer N60 instrument.

## cDNA synthesis and PCR of NMD targets

3 μg of total RNA were reverse transcribed in a 20 μl reaction volume with 10 μM VNN-(dT)20 primer using the GoScript Reverse Transcriptase (Promega) following the manufacturer's protocol. Synthesized cDNA was diluted in a ratio of 1:10. Thereof, 2% cDNA were used as template in end-point PCRs using the MyTaq Red Mix (Bioline) to study the abundance of NMD-sensitive targets. Sense and antisense primer were used at a final concentration of 0.2 μM (Supplementary Data 1). After 30 PCR cycles, the samples were separated by ethidium bromide-stained 1% agarose gels in 1x TB buffer at 140 V for 25 min. A 1 kb DNA ladder (Meridian Bioscience) was used as a size reference. DNA bands were visualized by trans-UV illumination using the Gel Doc XR+ (Bio-Rad) and quantified using the ImageLab software (Bio-Rad, version 6.0.1). The latter was also used for quantification of the detected bands. Uncropped scans of all gels are supplied in the source data file. Sanger sequencing of individual bands was performed using the service of Eurofins Genomics. For quantitative RT-PCR, 2% of cDNA and 0.2-0.6 μM of each primer (Supplementary Data 1) were used in 10 μl reactions on a CFX96 Touch Real-Time PCR Detection System with CFX Manager v3.0 software (BioRad). Each biological replicate was assayed in technical triplicates and the average Ct (threshold cycle) value was used for analysis. Mean log2 fold changes were calculated from three biologically independent experiments. Data are presented as individual values with the mean indicated. Raw data are supplied in the source data file.

## Protein extraction and Bradford assay

Cells were washed once with PBS and lysed in 100 μl ice-cold RIPA buffer (50 mM Tris/HCl pH 8.0, 0.1% SDS, 150 mM NaCl, 1% IGEPAL, 0.5% deoxycholate) supplemented with phosphatase inhibitors per well of a 6-well plate, followed by a 10 min incubation on ice. Lysates were transferred to 1.5 ml tubes and centrifuged at maximum speed for 10 min at 4 °C. Supernatants were used for protein quantification and subsequent western blot analysis.

Protein concentrations were determined by Pierce Detergent Compatible Bradford assay (Thermo Fisher Scientific). 5 μl of BSA standards or samples were mixed with 200 μl Bradford reagent in a 96-well plate. Absorbance at 595 nm was measured using an Apollo ELISA plate reader (Berthold), and concentrations were calculated from the standard curve.

## Western blotting

For SDS-PAGE, protein samples were mixed with 6x SDS loading buffer and resolved on 8% polyacrylamide gels in 1x SDS running buffer at 200 V for 50 min. PageRuler Plus Prestained Protein Ladder (Thermo Fisher Scientific) was used for molecular weight estimation. Proteins were transferred to Amersham Hybond P 0.2 PVDF membranes (Cytiva) by semi-dry blotting for 2 h at 18 V. Successful transfer was verified by stain-free imaging using the ChemiDoc XRS Imaging System (Bio-Rad). Membranes were blocked in 5% milk in TBS-T for 1 h at room temperature and incubated overnight at 4 °C with primary antibodies (Supplementary Data 1). Following three 5 min washes in TBS-T, membranes were incubated with HRP-conjugated secondary antibodies for 1 h at room temperature, washed twice for 5 min and once for 30 min to reduce background. Signals were detected using Amersham ECL Prime Western Blotting Detection Reagent (Cytiva) and visualized with the Fusion FX6 Edge imaging system (Vilber Lourmat). Uncropped scans of all blots are supplied in the source data file.

## RNA-seq library preparation and sequencing

For RNA-seq experiments, cells from three independent biological replicates per condition ($n = 3$) were lysed using 1 ml in-house prepared TRI reagent per 6 well[42]. The RNA was extracted and purified using the Direct-zol RNA MiniPrep kit including the recommended DNase I treatment (Zymo Research; Cat# R2052) according to manufacturer's instructions. ERCC RNA Spike-In Mix 1 (Thermo Fisher Scientific, Cat# 4456740) was added to the total RNA sample before library preparation. Libraries were prepared using the Stranded mRNA Preparation Kit (Illumina). Library preparation started with 500 ng total RNA. After poly-A selection (using poly-T oligo-attached magnetic beads), mRNA was purified and fragmented using divalent cations under elevated temperature. The RNA fragments underwent reverse transcription using random primers. This was followed by second strand cDNA synthesis. After end repair and A-tailing, indexing adapters were ligated. The products were then purified and amplified (12 PCR cycles) to create the final cDNA library. After validation (TapeStation, Agilent Technologies) and quantification (Qubit, Thermo Fisher Scientific) individual libraries were pooled. The library pools were quantified using the Collibri Library Quantification Kit (Thermo Fisher Scientific) and the QuantStudio 5 Real-Time PCR System (Thermo Fisher Scientific). Libraries were subsequently sequenced on an Illumina NovaSeq 6000 instrument using a $2 \times 100$ bp sequencing protocol and aiming for 50 million clusters per sample.

## Computational analyses of RNA-Seq data

Reads were aligned against the human genome (GRCh38, GENCODE release 42 transcript annotations[43] supplemented with SIRVomeERCCome annotations from Lexogen; obtained from https://www.lexogen.com/sirvs/download/) using the STAR read aligner (version 2.7.10b)[44]. Transcript abundance estimates were computed with Salmon (version 1.9.0)[45] in mapping-based mode using a decoy-aware transcriptome (GENCODE release 42 or NMDRHT version 1.2[36]) with the options --numGibbsSamples 30 --useVBOpt --gcBias --seqBias. After the import of transcript abundances in R (version 4.3.0) using tximport (version 1.28.0)[46], differential gene expression (DGE) analysis was performed with the DESeq2 R package (version 1.40.1)[47]. Genes with less than 10 counts in half the analyzed samples were pre-filtered and discarded. The DESeq2 log2FoldChange estimates were shrunk using the apeglm R package (version 1.22.1)[48]. P-values were calculated by DESeq2 using a two-sided Wald test and corrected for multiple testing using the Benjamini-Hochberg method. Differential transcript expression (DTE) analysis was performed using the edgeR R package (version 3.42.4)[49,50], accounting for mapping ambiguity by estimating the overdispersion based on 30 inferential replicate datasets drawn by Salmon using Gibbs sampling and subsequent count scaling. Lowly expressed transcripts were pre-filtered using the edgeR-internal

functions. P-values were calculated by edgeR using a quasi-likelihood F-test and corrected for multiple testing using the Benjamini-Hochberg method. General significance cutoffs, as long as not indicated otherwise, were log2FoldChange > 1 & padj <0.0001 for DESeq2 DGE and log2FC > 1 & FDR < 0.0001 for edgeR DTE. IGV snapshots were generated using igv.js[51]. Boxplots were generated using the geom_boxplot() function of ggplot2 with the centerline representing the 50th percentile (median), whereas the lower and upper box limits correspond to the 25th and 75th percentiles. The whiskers extend from the box limits to the smallest or largest value no further than 1.5 * interquartile range. Data beyond the end of the whiskers (outliers) are not shown.

## Crosslinking mass spectrometry

For intermolecular crosslinking with the amine-reactive $BS^3$ crosslinker, the purified proteins were mixed in a final concentration of 10 µM in crosslinking buffer (20 mM HEPES-KOH, pH 7.5; 50 mM KCl; 2 mM $MgCl_2$; 2 mM DTT) and incubated for 30 min at 30 °C to allow for complex formation. The crosslinking reaction was initiated by addition of bis(Sulfosuccinimidyl) suberat ($BS^3$, Thermo Scientific) in a final concentration of 0.5 mM and the mixtures were incubated for 30 min on ice, before the reaction was quenched with Tris/HCl (pH 7.5) in a final concentration of 25 mM. One biological replicate of the crosslinking reaction was subjected for analysis by mass spectrometry.

Crosslinked samples were lysed by 1:1 addition of 8 M urea in 50 mM Tris and sonication using a Bioruptor Plus system (Diogenode) ten times for 30 s at high intensity. For reduction and alkylation, 10 mM TCEP and 40 mM CAA were added and approximately 5 ug of peptide material was loaded on SPEC tips (strong cation exchange SAX material, home-made) and prepared according to[52]. Peptides were eluted with 1% FA from the SPEC tips and approximately 400 ng of peptide material was loaded on Evotips (Evosep).

Peptides were then eluted from the Evotips onto an Aurora Elite C18 column (15 cm × 75 µm, 1.7 µm particle size, IonOpticks) using the Evosep Eno HPLC system, employing the Whisper Zoom 20 samples per day (SPD) method. Mass spectrometry analysis was carried out on an Orbitrap Eclipse (ThermoFisher) equipped with a FAIMS Pro interface set to standard resolution, with compensation voltages of −50 – 60 V and −45 – 55 V, operated in data-dependent acquisition mode. Full MS scans were collected from m/z 350 to 1450 Th at a resolution of 60,000 (at m/z 200 Th). The top 15 most intense precursor ions were selected for fragmentation using stepped higher-energy C-trap dissociation (HCD) at normalized collision energies of 19, 27, and 35. MS2 spectra were acquired with a resolution of 30,000 (at m/z 200 Th) across a dynamic m/z range. Normalized automatic gain control (AGC) targets were set to 300% for MS1 and 100% for MS2, with a maximum injection time of 25 ms for MS1 and "auto" for MS2. Ions with a charge state of +2 were excluded to prioritize crosslinked precursors.

Raw data were processed using Proteome Discoverer (version 2.5.0.400), specifying Trypsin/P as the protease, and allowing up to two missed cleavages. The database search was performed against a FASTA file containing the sequences of the proteins under investigation. DSS/BS3 was set as the crosslinker. Cysteine carbamidomethylation was specified as a fixed modification, while methionine oxidation and protein N-terminal acetylation were set as dynamic modifications. Identifications were only accepted with a minimal score of 40 and a minimal Δ score of 4. Filtering at 1% false discovery rate (FDR) at CSM and cross-link levels were applied.

Crosslinks were visualized using the xiNET tool[53] and further analyzed using the XMAS extension[54] in USCF ChimeraX.

## AlphaFold structure predictions

Protein structure predictions were performed using AlphaFold *v* 2.3.1[23,24] and full databases with modeling set to "multimer". Truncated

sequences of human SMG5 (residues 788-1016; UniProt: Q9UPR3) and SMG6 (residues 153-1419; UniProt: Q86US8; isoform 1) proteins were used as input. AlphaFold Multimer runs yielded five models with five predictions per model (25 structure predictions in total).

Predictions of protein-RNA complexes were performed using AlphaFold 3[25] using truncated protein sequences of human SMG6 (residues 565–1419; UniProt: Q86US8, isoform 1) and SMG5 (residues 800–1016; UniProt: Q9UPR3) as well as an 11-mer RNA (5′- $U_3$-UGAAC-$U_3$ −3′) and divalent metal ions as input. The RNA sequence corresponds to one of the preferred pentameric cleavage motifs (UGAAC) of SMG6 identified in human cells[55] flanked by oligo-U sequences at both ends. Five models were obtained from each AlphaFold 3 run.

Resulting computational models were initially visualized and further analyzed for prediction convergence using UCSF ChimeraX $v1.6 – v1.9$[56,57]. For the predicted buried surface area between the two proteins, the mean of the calculations for each of the five models resulting from the AlphaFold 3 run was taken as a final estimate.

### Multiple sequence alignments

For multiple sequence alignments of metazoan homologs of human SMG5 and SMG6 proteins, the full-length sequences were aligned with Clustal Omega[58] using the default parameters and visualized using Jalview[59]. The following homologs of SMG5 were used: *H. sapiens* (UniProt: Q9UPR3), *X. tropicalis* (UniProt: Q05B16), *D. rerio* (UniProt: F1R7R1), *D. melanogaster* (UniProt: Q9V414), *C. elegans* (UniProt: G5ECF1). The following SMG6 homologs were used: *H. sapiens* (UniProt: Q86US8), *X. tropicalis* (UniProt: A0A6I8RBR9), *D. rerio* (UniProt: A0A0R4IZ84), *D. melanogaster* (UniProt: Q86BS8), *C. elegans* (UniProt: Q9BL68).

### Reporting summary

Further information on research design is available in the Nature Portfolio Reporting Summary linked to this article.

## Data availability

The data supporting the findings of this study are available from the corresponding authors upon request. Sequencing data generated in this study have been deposited at BioStudies/ArrayExpress under accession number E-MTAB-15521. Publicly available RNA sequencing datasets used in this study were obtained from E-MTAB-9330. Cross-linking MS data have been deposited to the ProteomeXchange Consortium via the PRIDE[60] partner repository with the dataset identifier PXD072588. Used published protein structures were PRORP1 from *A. thaliana* with PDB: 4G24. Raw data including original western blot, RT-PCR scans, RNA-seq analyses (including statistical parameters) and AlphaFold outputs have been deposited at Zenodo. Scoring metrics of AlphaFold runs are summarized in Supplementary Data 2. Source data are provided with this paper.

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

## Acknowledgements

We are grateful to Silke Modersohn and Fenja Meyer zu Altenschil-desche for technical assistance, as well as members of the Gehring lab for discussions (NHG lab); to Berenike Braun for her contributions in her Bachelor thesis work, to Petra Birle and Tatjana Krywcun for excellent technical assistance; to Fabien Bonneau, Jasemi Loukeri, Timo Reitinger, Marcela Cueto and Achim Keidel for valuable discussions (EC lab). We also thank Courtney Long for help in preparing the manuscript. For crosslinking data, we thank Barbara Steigenberger and the team at the MPIB Mass Spectrometry Facility (RRID:SCR_025745). N.H.G. acknowl-edges funding from the Deutsche Forschungsgemeinschaft (DFG, GE 2014/13-1; CRC 1678 project A08, grant agreement no. 520471345) and the Center for Molecular Medicine Cologne (CMMC) [project C 05]. E.C. acknowledges funding from the Max Planck Society, the European Research Council (Advanced Investigator Grant GOVERNA (101054447), the German Research Foundation SFB1035 (project A07), the Novo Nordisk Foundation ExoAdapt Grant (31199), the NOMIS Foundation and the Jung Foundation for Science and Research.

## Author contributions

Conceptualization: V.B., E.C., and N.H.G. Methodology: K.K., S.T., V.B., E.C., and N.H.G. Software and formal analysis: V.B. and K.K. Investiga-tion: K.K., S.T., and N.P. Resources and data curation: K.B. and V.B. Writing - original draft and review & editing: K.K., S.T., V.B., E.C., and N.H.G. Visualization: K.K., S.T., and V.B. Supervision and funding acqui-sition: E.C. and N.H.G.

## Funding

## Competing interests

The authors declare no competing interests.
