## [Transparent Peer Review file · Nature Communications]

Composite SMG5-SMG6 PIN domain formation is essential for NMD

Corresponding Author: Professor Niels Gehring

Version 0:

Reviewer comments:

Reviewer #1

(Remarks to the Author)

This manuscript provides the biggest advance in understanding the molecular mechanism of NMD in the last ten years and explains some longstanding enigmatic observations in the field. When SMG6 was discovered to be the endonuclease initiating NMD by cleaving nonsense mRNAs near the premature termination codon, it was puzzling that purified SMG6 PIN domains were so inefficient in cleaving circular RNA *in vitro*, suggesting that co-factors were missing (Eberle et al., 2009). Moreover, the finding that SMG5 was also required for the SMG6-mediated NMD pathway (Boehm et al., 2021) challenged the prevailing model of two independent branches of NMD, one mediated by SMG6 and the other by SMG5/SMG7. This exciting collaborative study between the Conti and the Gehring labs now documents that the fully active endonuclease is formed by a composite interface between the SMG5 and the SMG6 PIN domains.

First, the authors showed with knockdown/rescue experiments that in SMG5 KD cells, a SMG5 PIN mutant protein fails to restore the decay of the PTC-containing splice isoform of SRSF2 mRNA. Since this was in cells lacking SMG7, it indicated that the SMG5 PIN domain is required to promote SMG6-mediated NMD. Indeed, in *in vitro* nuclease assays, purified recombinant SMG5 PIN (800-1016) strongly stimulated the intrinsically weak activity of purified recombinant SMG6(565-1419), and both constructs weakly bound each other. AlphaFold models predicted with high confidence an interaction between the PIN domains of SMG6 and SMG5, and specific mutations in SMG5 and SMG6 predicted to interrupt this interaction indeed abolished the stimulation of nuclease activity of SMG6-PIN by SMG5-PIN *in vitro* and were unable to rescue NMD *in vivo*. The AF modeling further predicted SMG5 D893 to be part of the catalytic center of the endonuclease formed by the composite PIN domains of SMG6 and SMG5, which was also experimentally validated *in vitro* and *in vivo*. Moreover, two lysines of SMG5 (K896 and K897) predicted to interact with the RNA backbone were also shown to be required for the efficient nuclease activity, indicating that these positively charged residues are involved in correctly positioning the RNA for endonucleolytic cleavage.

Using combined siRNA- and dTAG-mediated depletion of SMG5 and SMG6 to achieve the strongest possible depletion of the respective protein in cells showed that SMG5 depletion targeted essentially the same transcripts as UPF1 depletion, and also the overlap with cells depleted of SMG6 was very high. These data demonstrate that UPF1, SMG5, and SMG6 are each indispensable for NMD.

Overall, this manuscript is very well written, the presented data is of high quality, and the conclusions are compelling and represent a substantial step forward in elucidating the mechanism of NMD in mammalian cells. This manuscript should be published as soon as possible. I only have a few minor points that can easily and quickly be addressed without requiring additional experiments.

Minor points:

Fig. 1E: It would be helpful to indicate in this cartoon which parts of the two proteins are present in FLAG-SMG5p and TS-SMG6tp.

Lines 143 and 272, ref 15: To the best of my knowledge, the Huntzinger et al., 2008 paper does not contain any *in vitro* nuclease activity assays. Therefore, this reference does not fit in this context.

Fig. 2C-D and 3C: In the labels on top of the gel images, it is difficult to see which black dots correspond to which of the constructs listed on the left. The longer the list is, the harder it becomes. I suggest adding fine lines that guide the eye from the construct name to the corresponding black dot.

Lines 229-236: While the dTAG degron system is elegant and induces a fast and efficient depletion of the targeted proteins, it is conceptually the same as an siRNA-mediated knockdown, because it also leaves residual amounts of proteins. It is therefore not logical to first point out the limitations of siRNA-mediated knockdowns and then write: "To overcome these constraints, we used the dTAG-1-inducible degron system". I suggest to first explain that knockouts are lethal and therefore impossible, then state that dTAG-mediated or siRNA-mediated depletions result only in partial NMD inhibition due to residual protein, and finally rationalizing the combined siRNA and dTAG knockdown as an attempt to minimize residual protein amounts and achieve more complete NMD inhibition.

Lines 241-242: In the same vein, UPF1 dTAG is a reference for a powerful/efficient but probably not complete NMD inhibition.

Fig 4D: The data shown here is difficult to comprehend in this non-standard and aggregated way of depicting it. More information about what exactly is shown in the left, middle, and right parts of the panels is needed in the figure legend for a faster and better understanding.

This review is from Oliver Mühlemann

Reviewer #2

(Remarks to the Author)

The present study by Kurscheidt et al. reports on a mechanistic investigation to understand how the endonucleolytic step of nonsense-mediated mRNA decay (NMD) is activated through direct cooperation between the proteins SMG5 and SMG6. Using a combination of structural modeling with AlphaFold, in vitro biochemical cleavage assays, and cell-based NMD assays, the authors demonstrate that the PIN domains of SMG5 and SMG6 form a weak but functional complex, that is called here composite PIN domain or cPIN, which assembles via a PIN-PIN heterodimeric interface. Structural predictions by AlphaFold suggested that SMG5 completes the catalytic and RNA-binding sites of SMG6. Subsequent biochemical assays confirmed that this composite SMG5-SMG6 complex efficiently cleaves RNA substrates, while mutational disruption of the predicted interface or RNA-binding regions affects nuclease activity in vitro and impairs NMD in cells. These findings redefine the current model of NMD by showing that SMG5 is not just a scaffold for the recruitment of decay factors by the SMG5-SMG7 pathway of NMD (deadenylation/decapping), but also an essential cofactor that directly activates SMG6-mediated endonucleolytic cleavage pathway of NMD.

The manuscript is well-written and the data support the conclusions and claims. Thus, the work is certainly suitable for publication since the provided data give a deeper understanding of the role of SMG5 at the crossroad of the two major NMD pathways.

I would recommend its publication once the following points have been addressed.

Major points:

1) Even though the models generated by AlphaFold (2 Multimer and 3) appear highly plausible based on subsequent biochemical and cell-based assays, it remains essential to report the relevant scoring metrics associated with these predictions. In particular, and for all modelling conducted with AlphaFold 2 Multimer and AlphaFold 3, one needs to report on the ipTM scores (for each pair of interacting partners, e.g. SMG5/SMG6, SMG6/RNA, SMG6/ions, etc.) that assess the confidence of complex formation at the interface, along with the predicted aligned error (pAE) matrices, which provides crucial indications of model reliability in the case of multi-component assemblies. In addition, displaying the SMG5/SMG6 complex with pLDDT-score colors is also important to report on the various degree of reliability of the different regions. Reporting these values (ipTM, pAE and pLDDT) represents good scientific practice and ensures accurate interpretation of complex models derived from AlphaFold.

2) For the biochemical in vitro cleavage assays, subtle differences in the cleavage patterns on gels (e.g., Figure 2C, lanes 2 and 3-4; Figure 2C, lanes 9 and 12-14; and Figure 2D, lanes 7 and 9) are overinterpreted, leading to conclusions that are not always convincing. These gels report single experiments, and the authors attempt to detect a gradation in the response to mutations. In my opinion, these experiments can be used to describe effects at a qualitative level (i.e., whether the mutant is active or not active) but analyses aiming to rank mutants by activity (e.g., this one being more active than that one) are problematic. Based solely on the reported gel images, one cannot reliably compare different levels of activity. If such quantitative interpretation is needed, independent repetitions of the experiments, accompanied by quantitative analysis, would be required. For instance, but not limited to, the differences between SMG6- Δ L-wt and SMG6-wt (Figure 2D), and between SMG6-I1/2/3 and SMG6-wt (Figure 2C), are not obvious (the differences look very similar). Yet, in one case, this is interpreted as residues being crucial for interaction and activity, while in the other (the Δ L deletion mutant), it is interpreted as this region being negligible for endonucleolytic activity. I recommend that the in vitro cleavage assays be re-analyzed and presented in a more objective manner. Repeats of the same cleavage assays, with quantitative analysis of the results, could be one option. Alternatively, focusing only on strong effects and interpreting them qualitatively could be another.

Minor points:

3) page 8: "Closer inspection of the computational models (Fig. 1E-F; Extended Data Fig. 1B-C) showed that the

intramolecular interaction within SMG6 involves the extended segment of the SMG6 PIN domain (res. 1239-1244 and 1365-1384) as well as part of the flexible linker connecting the 14-3-3-like domain with the helical hairpins domain in the TPR-like region." Is Fig. 1E-F appropriate here? Also, it is not easy to identify the the residues mentioned (res. 1239-1244 and 1365-1384), as well as the flexible linker mentioned, on the structural Figures, such as Fig. 1D. One should help the reader with additional Figures and or labels on the existing Figures to guide its lecture and help understanding the structural arrangement of SMG5/SMG6 interaction.

4) page 11: "Principal component analysis (PCA) showed that SMG5- and UPF1-depleted conditions clustered together, whereas SMG6-depleted samples formed a distinct group (Fig. 4C)." In the PCA analysis, PC1 and PC2 explain a large proportion of the variance (57% and 30%), but their biological or experimental meaning is not described. The authors should clarify what these components capture to aid interpretation of the results.

5) page 13: "Such division of labor, where one PIN domain contributes catalytic chemistry and RNA guidance while the other provides the primary active site, may represent a recurrent strategy in RNA surveillance machines." The authors propose a recurrent strategy, but as it is based on only one example and lacks references to other potential similar behaviors, this proposition seems somewhat far-fetched.

6) Discussion: In terms of molecular mechanism and sequential steps: Can SMG6 bind to RNA alone, or does it require SMG5? Can SMG6 bind Mg²⁺/Mn²⁺ ions alone, or does it require SMG5? Does SMG6/SMG5 need to be assembled first before binding RNA? Is there an order of events here that could be discussed?

7) page 19: "For proteins used in in vitro nuclease assays, protein purification was carried out in an RNase-free manner." What is meant exactly by an RNase-free manner? What is done differently for the protein purifications, depending on whether they are used in nuclease assays or not? Usually, RNase contamination comes from E. coli protein contaminants, and it is not easy to simply decide being in an RNase-free versus a standard protein purification procedure.

8) page 20: "20 nM of fluorescently labelled substrate was mixed with 3.6 μM of each protein (180x excess of protein over substrate)." One might need to comment in the main text the unusual experimental conditions for an enzymatic activity assay, with an 180x excess of enzyme over substrate.

9) page 27: "as well as an 11-mer RNA (5'- U4-GAAC-U3 -3')". How the RNA sequence was chosen and why? Does it make a difference if one select other RNA sequences? Which nucleotides are positioned to be cleaved by SMG6? These points are worth mentioning in the main text.

10) Figure1: What is the buried surface area of the complex between SMG5 and SMG6? This is an informative figure that is worth mentioning in the text.

Reviewer #3

(Remarks to the Author)

Degradation of NMD-targeted transcripts in metazoans has long been thought to be carried out by two independent and partially redundant pathways, one mediated by SMG6 for endonucleolytic cleavage and another mediated by SMG5-SMG7 for deadenylation-dependent decapping. Contrary to this view, recent experiments from the Gehring lab indicate that SMG5, SMG6, and SMG7 collaborate with each other and appear to function in the same decay pathway. In the current manuscript, the authors address the possibility of functional collaboration of SMG5 and SMG6 PIN domains during the activation of mammalian NMD. Using structural predictions, in vitro biochemical assays, and cell-based NMD analyses, they provide results indicating that: 1) SMG5 and SMG6 PIN domains interact and appear to form a composite catalytic site with enhanced endonuclease activity, and 2) disruptions of the predicted SMG5-SMG6 PIN domain-domain interactions or putative SMG5 PIN domain interaction with RNA essentially eliminate NMD of selected substrates in vivo. Based on these results, the authors propose that SMG5 and SMG6 form a composite PIN domain of high degradation efficiency, with the SMG5 PIN domain completing the SMG6 active site and providing additional RNA substrate binding and catalytic functions. This would be an important conclusion and could provide significant insights into the structure, function, and mechanism of action of SMG5 and SMG6 during activation of NMD. However, the data presented in the manuscript are not convincing and additional experiments are needed to solidify conclusions regarding the functional collaboration of SMG5 and SMG6 during mammalian NMD.

Major concerns

1) In all biochemical assays of endonucleolytic activity, the purified Flag-SMG5p-wt (Figure 1H-lane 9, Figure 2C-lane 8, Figure 2D-lane 6, and Figure 3C-lane 4) was claimed to have no activity but did exhibit endonuclease activity comparable to that of purified TS-SMG6TP-wt (Figure 1H-lane 5, Figure 2C-lane 2, Figure 2D-lane 2, and Figure 3C-lane 2). Thus, the enhanced endonucleolytic cleavage activities of TS-SMG6TP-wt (Figure 1H-lanes 11-13) or some of SMG6 mutants (Figure 2C-lanes 12-13, Figure 2D-lane 9) observed by adding Flag-Smg5p-wt may not have resulted from interactions of the SMG5-SMG6 PIN domains, but from independent activities of individual domains. In fact, purified Flag-SMG5—C1, R1, I3 (Figure 3C-lanes 5, 6, and 8) also exhibited significant cleavage activity. This raises the possibility that purified SMG5 fragments were contaminated by endo or exonucleases and question the validity of the conclusion "direct SMG5-SMG6 interactions stimulate endonucleolytic activity in vitro."

2) To strengthen the conclusion that SMG5 and SMG6 form a composite PIN domain essential for NMD, the effects of

different SMG5 and SMG6 mutants should be evaluated either in vitro or in vivo. Examples of the types of mutants that ought to be examined are those that specifically disrupt postulated protein-protein or RNA-protein interactions or postulated catalytic or RNA binding functions. These mutants, in the context of full-length proteins, should also be evaluated for in vivo NMD activities with bona fide NMD substrates, i.e., preferably not the transcripts currently analyzed in this manuscript including SRSF2 (Figures 1C, 2B, 3B, 4B, Extended Data Figure 4A), RBM3 (Extended Data Figures 1A, 2C, 3B) and SRSF6 (Extended Data Figures 1A, 2D, 3C, 4B). These transcripts all exhibit isoform switches to different extents and the increased accumulation of the PTC+ isoforms observed in NMD-deficient cells may only arise partially from direct effects of NMD loss. Analyses of bona fide NMD substrates in different SMG5 and SMG6 mutants have the potential to detect different decay intermediates and thus facilitate functional assignments for different residues in both SMG5 and SMG6.

3) In the SMG5 and SMG6 function rescue experiments (Figures 3B and 2B), the integrated exogenous SMG5 or SMG6 mutant alleles are highly over-expressed compared to that from their respective endogenous alleles (Figures 3B and 2B, both lane1, bands invisible). The apparent loss of NMD function for some of the SMG5 and SMG6 mutants may thus originate from dominant-negative inhibition. Some of these mutants should be tested in WT cells for dominant-negative activity.

4) AlphaFold structure prediction is a powerful tool. To evaluate the prediction results, it will be helpful if the authors provided some confidence measures for the structure predictions presented in this manuscript including Figures 1D, 2A, 3A, and Extended Data Figures 1B, 1C, 2A, 2B, 3A).

Version 1:

Reviewer comments:

Reviewer #2

(Remarks to the Author)

I am overall satisfied with the revised version and with the fact that the authors have addressed all my comments carefully. The authors have clearly improved their manuscript during the revision, in response to the request of the three reviewers. I strongly recommend it for publication.

Reviewer #3

(Remarks to the Author)

The authors have done an excellent job of addressing the comments raised in my review as well as those raised by the other two reviewers. The results of this paper provide considerable and long sought insights into the mechanism of metazoan NMD and, as such render this a highly significant and important paper. I recommend publication without any further revision.

REVIEWER COMMENTS

Reviewer #1 (Remarks to the Author):

This manuscript provides the biggest advance in understanding the molecular mechanism of NMD in the last ten years and explains some longstanding enigmatic observations in the field. When SMG6 was discovered to be the endonuclease initiating NMD by cleaving nonsense mRNAs near the premature termination codon, it was puzzling that purified SMG6 PIN domains were so inefficient in cleaving circular RNA in vitro, suggesting that co-factors were missing (Eberle et al., 2009). Moreover, the finding that SMG5 was also required for the SMG6-mediated NMD pathway (Boehm et al., 2021) challenged the prevailing model of two independent branches of NMD, one mediated by SMG6 and the other by SMG5/SMG7. This exciting collaborative study between the Conti and the Gehring labs now documents that the fully active endonuclease is formed by a composite interface between the SMG5 and the SMG6 PIN domains.

First, the authors showed with knockdown/rescue experiments that in SMG5 KD cells, a SMG5 Δ PIN mutant protein fails to restore the decay of the PTC-containing splice isoform of SRSF2 mRNA. Since this was in cells lacking SMG7, it indicated that the SMG5 PIN domain is required to promote SMG6-mediated NMD. Indeed, in in vitro nuclease assays, purified recombinant SMG5 PIN (800-1016) strongly stimulated the intrinsically weak activity of purified recombinant SMG6(565-1419), and both constructs weakly bound each other. AlphaFold models predicted with high confidence an interaction between the PIN domains of SMG6 and SMG5, and specific mutations in SMG5 and SMG6 predicted to interrupt this interaction indeed abolished the stimulation of nuclease activity of SMG6-PIN by SMG5-PIN in vitro and were unable to rescue NMD in vivo. The AF modeling further predicted SMG5 D893 to be part of the catalytic center of the endonuclease formed by the composite PIN domains of SMG6 and SMG5, which was also experimentally validated in vitro and in vivo.

Moreover, two lysines of SMG5 (K896 and K897) predicted to interact with the RNA backbone were also shown to be required for the efficient nuclease activity, indicating that these positively charged residues are involved in correctly positioning the RNA for endonucleolytic cleavage.

Using combined siRNA- and dTAG-mediated depletion of SMG5 and SMG6 to achieve the strongest possible depletion of the respective protein in cells showed that SMG5 depletion targeted essentially the same transcripts as UPF1 depletion, and also the overlap with cells depleted of SMG6 was very high. These data demonstrate that UPF1, SMG5, and SMG6 are each indispensable for NMD.

Overall, this manuscript is very well written, the presented data is of high quality, and the conclusions are compelling and represent a substantial step forward in elucidating the mechanism of NMD in mammalian cells. This manuscript should be published as soon as possible. I only have a few minor points that can easily and quickly be addressed without requiring additional experiments.

Minor points:

Fig. 1E: It would be helpful to indicate in this cartoon which parts of the two proteins are present in FLAG-SMG5p and TS-SMG6tp.

The parts of the molecules used in the in vitro assays (FLAG-SMG5_p and TS-SMG6_{TP}) are now highlighted in the cartoon (Fig. 1E) with dashed rectangular boxes.

Lines 143 and 272, ref 15: To the best of my knowledge, the Huntzinger et al., 2008 paper does not contain any in vitro nuclease activity assays. Therefore, this reference does not fit in this context.

We have removed this reference at both locations.

Fig. 2C-D and 3C: In the labels on top of the gel images, it is difficult to see which black dots correspond to which of the constructs listed on the left. The longer the list is, the harder it becomes. I suggest adding fine lines that guide the eye from the construct name to the corresponding black dot.

We added fine lines to help the reader trace each construct name to its corresponding black dot.

Lines 229-236: While the dTAG degron system is elegant and induces a fast and efficient depletion of the targeted proteins, it is conceptually the same as an siRNA-mediated knockdown, because it also leaves residual amounts of proteins. It is therefore not logical to first point out the limitations of siRNA-mediated knockdowns and then write: "To overcome these constraints, we used the dTAG-1-inducible degron system". I suggest to first explain that knockouts are lethal and therefore impossible, then state that dTAG-mediated or siRNA-mediated depletions result only in partial NMD inhibition due to residual protein, and finally rationalizing the combined siRNA and dTAG knockdown as an attempt to minimize residual protein amounts and achieve more complete NMD inhibition.

We have moved this part of the manuscript to an earlier position in the Results section and rephrased it in accordance with the reviewer's comments. The revised text can now be found in lines 168-180:

"Having established that SMG5 directly enhances the catalytic activity of the SMG6 PIN domain in vitro, we next asked whether this dependency is reflected in cellular NMD. Despite long-standing assumptions regarding the essentiality of SMG5 and SMG6, it has remained unclear whether NMD can proceed under near-complete depletion of either factor, in part because siRNA-mediated depletion leaves residual protein sufficient to sustain partial pathway activity. Moreover, the essentiality of SMG5 and SMG6 in cells has precluded the generation of conventional knockout cell lines. To circumvent these limitations, we engineered conditional depletion lines in which endogenous SMG5 or SMG6 was fused to FKBP12F36V, enabling their rapid and inducible degradation using the dTAGV-1 degron system (Fig. 2A). Because both siRNA-mediated and degron-mediated protein depletion typically leave residual protein that limits the extent of NMD inhibition (Extended Data Fig. 4A, lanes 3,4,6,9), we combined dTAG-mediated degradation with siRNA-mediated knockdown, enabling a more complete inhibition of NMD (Fig. 2B; Extended Data Fig. 4A-B)."

Lines 241-242: In the same vein, UPF1 dTAG is a reference for a powerful/efficient but probably not complete NMD inhibition.

We now describe the UPF1 dTAG system as providing "a very efficient NMD inhibition" (line 181-182).

Fig 4D: The data shown here is difficult to comprehend in this non-standard and aggregated way of depicting it. More information about what exactly is shown in the left, middle, and right parts of the panels is needed in the figure legend for a faster and better understanding.

We have replaced Fig 4D with a less complex depiction of transcript-level analysis. Since the original analysis is still included in the supplemental figures (including other RNA-seq data), we nevertheless expanded the corresponding figure legend to aid a faster and better understanding.

Reviewer #2 (Remarks to the Author):

The present study by Kurscheidt et al. reports on a mechanistic investigation to understand how the endonucleolytic step of nonsense-mediated mRNA decay (NMD) is activated through direct cooperation between the proteins SMG5 and SMG6. Using a combination of structural modeling with AlphaFold, in vitro biochemical cleavage assays, and cell-based NMD assays, the authors demonstrate that the PIN domains of SMG5 and SMG6 form a weak but functional complex, that is called here composite PIN domain or cPIN, which assembles via a PIN-PIN heterodimeric interface. Structural predictions by AlphaFold suggested that SMG5 completes the catalytic and RNA-binding sites of SMG6. Subsequent biochemical assays confirmed that this composite SMG5–SMG6 complex efficiently cleaves RNA substrates, while mutational disruption of the predicted interface or RNA-binding regions affects nuclease activity in vitro and impairs NMD in cells. These findings redefine the current model of NMD by showing that SMG5 is not just a scaffold for the recruitment of decay factors by the SMG5-SMG7 pathway of NMD (deadenylation/decapping), but also an essential cofactor that directly activates SMG6-mediated endonucleolytic cleavage pathway of NMD.

The manuscript is well-written and the data support the conclusions and claims. Thus, the work is certainly suitable for publication since the provided data give a deeper understanding of the role of SMG5 at the crossroad of the two major NMD pathways.

I would recommend its publication once the following points have been addressed.

Major points:

1) Even though the models generated by AlphaFold (2 Multimer and 3) appear highly plausible based on subsequent biochemical and cell-based assays, it remains essential to report the relevant scoring metrics associated with these predictions. In particular, and for all modelling conducted with AlphaFold 2 Multimer and AlphaFold 3, one needs to report on the ipTM scores (for each pair of interacting partners, e.g. SMG5/SMG6, SMG6/RNA, SMG6/ions, etc.) that assess the confidence of complex formation at the interface, along with the predicted aligned error (pAE) matrices, which provides crucial indications of model reliability in the case of multi-component assemblies. In addition, displaying the SMG5/SMG6 complex with pLDDT-score colors is also important to report on the various degree of reliability of the different regions. Reporting these values (ipTM, pAE and pLDDT) represents good scientific practice and ensures accurate interpretation of complex models derived from AlphaFold.

We agree that the scoring metrics and matrices are important to assess the quality of the predictions. We now provide per-residue pLDDT plots for all models as well as a PAE matrix for

one model of each AlphaFold run. Furthermore, we display this model coloured by pLDDT score. To accommodate these additions, we now display the overlay of all models as well as the corresponding scoring metrics and matrices of the AlphaFold Multimer and AlphaFold 3 runs in Extended Data Fig. 2 and 3, respectively. Furthermore, we summarized the predicted template modelling (pTM) and interface predicted template modelling (ipTM) scores as well as other scoring metrics of our predictions in an additional table (Table 2).

2) For the biochemical in vitro cleavage assays, subtle differences in the cleavage patterns on gels (e.g., Figure 2C, lanes 2 and 3–4; Figure 2C, lanes 9 and 12–14; and Figure 2D, lanes 7 and 9) are overinterpreted, leading to conclusions that are not always convincing. These gels report single experiments, and the authors attempt to detect a gradation in the response to mutations. In my opinion, these experiments can be used to describe effects at a qualitative level (i.e., whether the mutant is active or not active) but analyses aiming to rank mutants by activity (e.g., this one being more active than that one) are problematic. Based solely on the reported gel images, one cannot reliably compare different levels of activity. If such quantitative interpretation is needed, independent repetitions of the experiments, accompanied by quantitative analysis, would be required. For instance, but not limited to, the differences between SMG6- Δ L-wt and SMG6-wt (Figure 2D), and between SMG6-I1/2/3 and SMG6-wt (Figure 2C), are not obvious (the differences look very similar). Yet, in one case, this is interpreted as residues being crucial for interaction and activity, while in the other (the Δ L deletion mutant), it is interpreted as this region being negligible for endonucleolytic activity. I recommend that the in vitro cleavage assays be re-analyzed and presented in a more objective manner. Repeats of the same cleavage assays, with quantitative analysis of the results, could be one option. Alternatively, focusing only on strong effects and interpreting them qualitatively could be another.

Although we show a single gel, this is a representative of at least three independent experiments, all showing the same pattern. However, we agree with the Reviewer and have revised the text to focus primarily on the strongest effects, while describing the others qualitatively. The gels from the in vitro experiments using the different mutant constructs are attached to this response letter.

In the manuscript, we now state (lines 221-228 and 236-243: “All three interaction mutants of SMG6 (SMG6 I1-I3) showed defects in NMD rescue activity when expressed in SMG6 degron cells, with I3 being most impaired, I1 moderately affected, and I2 retaining the most activity (Fig. 3D, lanes 7-9; Extended Data Fig. 5C-D, lanes 7-9). In nuclease assays in vitro, the SMG6 I3 mutation impaired the stimulation of SMG6 catalytic activity by SMG5 (Fig. 3E, lane 14), while it had no effect on the activity of SMG6 alone (Fig. 3E, lane 7). Qualitatively, the I1 and I2 mutants had weaker yet reproducible effects on the stimulated catalytic activity in vitro (Fig. 3E, lanes 12-13), consistent with their cellular NMD phenotypes.

...

We assessed the impact of the intramolecular interaction between the loop region in the TPR-like helical domain and the extended β -sheet region in the PIN domain of SMG6 using a loop deletion construct (SMG6 Δ L; deletion of residues 845-868). Although the effect on the catalytic activity in vitro was not pronounced (Fig. 3F), the loop deletion substantially impaired NMD in rescue assays in vivo (Fig. 3D, lane 6; Extended Data Fig. 5C-D, lane 6). Thus, the loop is not

required for catalytic activity per se but is important for SMG6 function in cells, in line with a proposed function of the extended segment of the SMG6 PIN domain in correctly positioning the PIN domain with respect to other domains of SMG6."

In vitro activity assays with SMG6 mutants and SMG6 loop-deletion constructs:

In vitro activity assays with SMG5 mutants:

A

SMG5 mutants	residues
TS55A	55
TS55R	55
K95D	95
K95E	95
K95G	95
K95H	95
K95I	95
K95L	95
K95M	95
K95N	95
K95Q	95
K95R	95
K95S	95
K95T	95
K95V	95
K95W	95
K95Y	95
K95D	95
K95E	95
K95G	95
K95H	95
K95I	95
K95L	95
K95M	95
K95N	95
K95Q	95
K95R	95
K95S	95
K95T	95
K95V	95
K95W	95
K95Y	95

Minor points:

3) page 8: "Closer inspection of the computational models (Fig. 1E-F; Extended Data Fig. 1B-C) showed that the intramolecular interaction within SMG6 involves the extended segment of the SMG6 PIN domain (res. 1239-1244 and 1365-1384) as well as part of the flexible linker connecting the 14-3-3-like domain with the helical hairpins domain in the TPR-like region." Is Fig. 1E-F appropriate here? Also, it is not easy to identify the residues mentioned (res. 1239-1244 and 1365-1384), as well as the flexible linker mentioned, on the structural Figures, such as Fig. 1D. One should help the reader with additional Figures and or labels on the existing Figures to guide its lecture and help understanding the structural arrangement of SMG5/SMG6 interaction.

Indeed, we should have referred to Fig. 1D-E (instead of 1E-F). The text has been corrected.

Following the Reviewer's comment, we added an additional schematic summarizing the catalytic as well as the mutated residues (Fig. 3A). This schematic also includes the deletion mutant of the flexible linker in the SMG6 TPR-like domain.

4) page 11: "Principal component analysis (PCA) showed that SMG5- and UPF1-depleted conditions clustered together, whereas SMG6-depleted samples formed a distinct group (Fig. 4C)." In the PCA analysis, PC1 and PC2 explain a large proportion of the variance (57% and 30%), but their biological or experimental meaning is not described. The authors should clarify what these components capture to aid interpretation of the results.

We have analyzed the PCA results in more detail and determined the "loadings" of each gene, which allowed to estimate the contribution of individual gene expression variances to the principle components 1 and 2 (see Panel A; Figure below). We selected the top 100 genes (ranked by absolute loading) for PC1 & PC2 and found that the effects captured in PC1 are mostly related to NMD. To do so, we used a previously calculated "NMD relevance" score (Boehm et al. 2025), which determines the significant up-/downregulation in up to 18 NMD-compromised conditions and we found that the top 100 PC1 genes have high NMD scores (Panel B; Figure below; also now included in main Figure 4C), whereas PC2 genes showed substantially lower scores.

Next, we visualized the differential gene expression profiles of the top 100 PC1 & PC2 genes (Panel C; Figure below). Most PC1 genes positively contributed to the separation along PC1 and showed increased gene expression upon NMD inhibition. Interestingly, most of the top 100 PC1 genes are rapidly upregulated upon UPF1 depletion (early or delayed DGE cluster; Boehm et al. 2025), which further substantiates that these are NMD-related effects.

In contrast, the majority of the top 100 PC2 genes are downregulated (negatively related to PC2) and do not belong to particular DGE clusters, indicating rather NMD-independent effects. We re-analyzed various published and unpublished RNA-seq data and found strikingly similar expression patterns for PC2 top 100 genes when SMG6 was downregulated via siRNA-mediated knockdown, either alone or in the SMG7 KO background (Panel C; Figure below). Thus, the gene expression changes contributing to PC2 are most likely SMG6-specific effects, either due to non-NMD functions of SMG6 (e.g. PMID: 12676087, <https://pubmed.ncbi.nlm.nih.gov/12676087/>) or because of off-target effects from the siRNA treatment. Gene list functional enrichment analysis of PC1 and PC2 genes did not reveal particularly insightful additional information. The RNA-seq coverage for the top "loading" gene for PC1 and PC2 was visualized in Panel D to provide examples for the gene expression profiles shown in Panel C.

5) page 13: "Such division of labor, where one PIN domain contributes catalytic chemistry and RNA guidance while the other provides the primary active site, may represent a recurrent strategy in RNA

surveillance machines." The authors propose a recurrent strategy, but as it is based on only one example and lacks references to other potential similar behaviors, this proposition seems somewhat far-fetched.

We agree that our conclusion requires additional explanation. Dimer formation has previously been observed for members of the PIN-domain-like superfamily, particularly in bacteria and archaea (Senissar ... Brodersen 2017 Prot Sci, <https://pubmed.ncbi.nlm.nih.gov/28508407/>). Furthermore, it has been reported that the homodimerization of the PIN domain of Regnase-1 from *M. musculus* is required for the nucleolytic activity of the enzyme in vitro (Yokogawa ... Inagaki 2016 Sci Rep, <https://pubmed.ncbi.nlm.nih.gov/26927947/>). To the best of our knowledge, there are currently no reports on PIN-PIN heterodimers, as it seems to be formed by SMG5 and SMG6. Both active and inactive PIN domains are, however, present in other pathways of RNA turnover and quality control, such as the PIN domain of Rrp44/DIS3 of the exosome, the inactive PIN domain of Nmd4 in yeast NMD and the PIN domain of Nob1 in rRNA processing, and some of the active nucleases have reported to exhibit weak catalytic activity in vitro. It is thus tempting to speculate that the formation of heterodimers of inactive and weakly active PIN domains could be a recurring mechanism in RNA surveillance. However, we acknowledge that future studies are required to support this claim.

Following up on the Reviewer's comment we added the above-mentioned examples of PIN-domain containing proteins involved in other pathways of RNA turnover and surveillance to support our tentative conclusion of a potential recurrent mechanism.

In the manuscript, we now state (lines 344-352): "Beyond the specific case of SMG5 and SMG6, the composite PIN module they form exemplifies how paralogous nuclease folds can cooperate to optimize catalysis. Such division of labor, where one PIN domain contributes catalytic chemistry and RNA positioning while the other provides the primary active site, may represent a recurrent strategy in RNA surveillance machines, considering that both active and inactive PIN domains are found in various complexes in RNA processing, turnover and quality control (such as in Nmd4, Nob1 and Rrp44/DIS3). It also offers a conceptual framework for understanding how transient assemblies can produce switch-like enzymatic outputs in cellular pathways."

6) Discussion: In terms of molecular mechanism and sequential steps: Can SMG6 bind to RNA alone, or does it require SMG5? Can SMG6 bind Mg²⁺/Mn²⁺ ions alone, or does it require SMG5? Does SMG6/SMG5 need to be assembled first before binding RNA? Is there an order of events here that could be discussed?

Since SMG6 alone retains weak catalytic activity in vitro, it seems possible that the protein can bind to RNA and also coordinate the divalent metal ions required for catalysis in absence of SMG5. Based on the strong effect of the SMG5 active-site mutant D893A as well as of the putative SMG5 RNA-binding mutant K896D K897D (R1), we would expect that both the metal coordination as well as the substrate binding of SMG6 are considerably enhanced in the presence of the SMG5 PIN domain. This being said, we would like to point out that our current data does not allow us to draw conclusions on the exact molecular mechanisms of the endonucleolytic cleavage carried out by SMG5-SMG6 and the order of events leading to the assembly of this complex.

It would, however, be interesting to address these aspects in the future and we hope that our work here can provide the basis for follow-up studies.

7) page 19: "For proteins used in in vitro nuclease assays, protein purification was carried out in an RNase-free manner." What is meant exactly by an RNase-free manner? What is done differently for the protein purifications, depending on whether they are used in nuclease assays or not? Usually, RNase contamination comes from E. coli protein contaminants, and it is not easy to simply decide being in an RNase-free versus a standard protein purification procedure.

We understand that the purification procedures require additional explanations. We have included the specific steps and clarifications in the purification details in the methods: The wash step with ATP and a high salt concentration (Buffer B) was included with the goal to remove co-purifying chaperones as well as nucleic acids and contaminating RNases. To reduce the possibility of introducing RNases into the protein preparations, the purification of constructs used in in vitro nuclease assays was carried out using RNase-free buffers and reagents as well as sterilized reaction tubes and the gel filtration columns and the connected injection loops were cleaned with sodium hydroxide prior to injection of the concentrated eluate from the affinity step.

In the manuscript we now state (lines 545-569 and 586-588): "TwinStrep-3C-tagged SMG6 fragments were recombinantly expressed in E. coli Star (DE3) pRARE. Cultures were grown to an optical density at 600 nm (OD600) of 1.0-1.5 at 37°C and 180 rpm, before the temperature was shifted to 18°C and protein expression was induced by addition of 0.5 mM IPTG. After incubation over-night, cells were harvested by centrifugation at 8,983 x g and resuspended in Buffer A (50 mM HEPES-KOH, pH 7.5; 500 mM NaCl; 2 mM DTT) supplemented with 1 mM AEBSF (PanReac AppliChem), DNase I (Roche), Benzonase (Merck), 2 mM MgCl₂, BioLock (IBA Lifesciences), and 0.1 mg/mL lysozyme. The cell suspension was lysed by ultrasonication and the cell debris was pelleted by centrifugation at 75,600 x g and 4°C for 30 min. The cleared lysate was passed over a 5-mL StrepTrap XT column (Cytiva) pre-equilibrated with Buffer A for affinity chromatography.

After The column was washed with Buffer A and Buffer B (20 mM HEPES-KOH, pH 7.5; 50 mM KCl; 1000 mM NaCl; 10 mM MgSO₄; 2 mM ATP; 10% (v/v) glycerol) and again Buffer A, before bound proteins were eluted with Buffer A supplemented with 50 mM biotin. The wash step with ATP and a high salt concentration (Buffer B) was included with the goal to remove co-purifying chaperones as well as nucleic acids and contaminating RNases. The eluate was subsequently concentrated using Amicon Ultra-15 centrifugal filters (Merck) with a molecular weight cut-off of 30 kDa and subjected to size-exclusion chromatography over a HiLoad 16/600 Superdex 200 pg column (Cytiva) in gel filtration buffer (20 mM HEPES-KOH, pH 7.5; 150 mM NaCl; 1 mM DTT). Peak fractions containing the protein of interest were pooled and concentrated using Amicon Ultra-4 centrifugal filters (Merck) with a molecular weight cut-off of 30 kDa. The concentrated protein was flash-frozen in liquid nitrogen and stored at -80°C until use. To reduce the possibility of introducing RNases into the protein preparations, the purification of constructs used in in vitro nuclease assays was carried out using RNase-free buffers and reagents as well as sterilized reaction tubes and the gel filtration columns and the connected injection loops were cleaned with sodium hydroxide prior to injection of the concentrated eluate from the affinity step.
SMG5 fragments...

For SMG5 constructs used in in vitro nuclease assays, similar precautions against introducing RNase contaminations were taken as described for the SMG6 constructs.

8) page 20: "20 nM of fluorescently labelled substrate was mixed with 3.6 μ M of each protein (180x excess of protein over substrate)." One might need to comment in the main text the unusual experimental conditions for an enzymatic activity assay, with an 180x excess of enzyme over substrate.

While we agree with the Reviewer that 180x constitutes a large excess of enzyme over substrate, we would like to point out that in vitro nuclease activity assays using a similarly large excess of enzyme over RNA substrate have been reported for other PIN domain-containing proteins, e.g. Swt1 from *S. cerevisiae* (Skruzny ... Tollervey, Hurt 2009 PloS Biology, <https://pmc.ncbi.nlm.nih.gov/articles/PMC2613419/>) and Utp24 from *S. cerevisiae* (Wells ... Schneider 2016 NAR, <https://pubmed.ncbi.nlm.nih.gov/27418679/>). Following the Reviewer's suggestion, we are now also commenting on the experimental conditions in the main text.

In the manuscript we now state (lines 127-132): "Similar to previous reports, TS-SMG6_{TP} alone only exhibited weak nucleolytic activity in vitro, which was even further impaired upon mutation of an active site residue (D1353A; Fig. 1H, lanes 3-5,7; Extended Data Fig. 1E). Owing to the weak nucleolytic activity of SMG6, we used a higher protein-to-RNA ratio that is consistent with conditions routinely applied to other eukaryotic PIN domains in vitro.

9) page 27: "as well as an 11-mer RNA (5'- U4-GAAC-U3 -3')". How the RNA sequence was chosen and why? Does it make a difference if one selects other RNA sequences? Which nucleotides are positioned to be cleaved by SMG6? These points are worth mentioning in the main text.

We chose the sequence based on a previous study, reporting that UGAAC is apparently one of the preferred pentameric cleavage motifs for SMG6 (Schmidt ... and Green 2015 NAR, <https://pubmed.ncbi.nlm.nih.gov/25429978/>). We have also included this information in the methods section.

From the computational models, the phosphate backbone of the RNA appears to be the major contact site. As such, we would expect a minor contribution from the specific RNA bases, and indeed AlphaFold 3 runs with different RNA sequences result in a similar position and coordination of the RNA by the two divalent metal ions in the active site. However, the computational model does suggest a preferred directionality with how the RNA chain binds the PIN-PIN interface.

In the manuscript we state (lines 814-819): "Predictions of protein-RNA complexes were performed using AlphaFold 3 using truncated protein sequences of human SMG6 (residues 565-1419; UniProt: Q86US8, isoform 1) and SMG5 (residues 800-1016; UniProt: Q9UPR3) as well as an 11-mer RNA (5'- U3-UGAAC-U3 -3') and divalent metal ions as input. The RNA sequence corresponds to one of the preferred pentameric cleavage motifs (UGAAC) of SMG6 identified in human cells flanked by oligo-U sequences at both ends. Five models were obtained from each AlphaFold 3 run."

10) Figure1: What is the buried surface area of the complex between SMG5 and SMG6? This is an informative figure that is worth mentioning in the text.

We calculated the buried surface area of the SMG5-SMG6 complex predicted by AlphaFold3 using the mean of the five models. The mean buried surface area at the immediate SMG5 PIN-SMG6 PIN interface amounts to 677 Å². We have also added this information to the main text.

In the manuscript we state (lines 92-95): “Computational predictions using AlphaFold Multimer or AlphaFold 3 consistently positioned the PIN-like folds of SMG6 and SMG5 in an extended side-by-side interaction burying a mean predicted surface area of 677 Å² at the immediate PIN-PIN interface (Fig. 1D, Extended Data Fig. 2A).”

Reviewer #3 (Remarks to the Author):

Degradation of NMD-targeted transcripts in metazoans has long been thought to be carried out by two independent and partially redundant pathways, one mediated by SMG6 for endonucleolytic cleavage and another mediated by SMG5-SMG7 for deadenylation-dependent decapping. Contrary to this view, recent experiments from the Gehring lab indicate that SMG5, SMG6, and SMG7 collaborate with each other and appear to function in the same decay pathway. In the current manuscript, the authors address the possibility of functional collaboration of SMG5 and SMG6 PIN domains during the activation of mammalian NMD. Using structural predictions, in vitro biochemical assays, and cell-based NMD analyses, they provide results indicating that: 1) SMG5 and SMG6 PIN domains interact and appear to form a composite catalytic site with enhanced endonuclease activity, and 2) disruptions of the predicted SMG5-SMG6 PIN domain-domain interactions or putative SMG5 PIN domain interaction with RNA essentially eliminate NMD of selected substrates in vivo. Based on these results, the authors propose that SMG5 and SMG6 form a composite PIN domain of high degradation efficiency, with the SMG5 PIN domain completing the SMG6 active site and providing additional RNA substrate binding and catalytic functions. This would be an important conclusion and could provide significant insights into the structure, function, and mechanism of action of SMG5 and SMG6 during activation of NMD. However, the data presented in the manuscript are not convincing and additional experiments are needed to solidify conclusions regarding the functional collaboration of SMG5 and SMG6 during mammalian NMD.

Major concerns

1) In all biochemical assays of endonucleolytic activity, the purified Flag-SMG5p-wt (Figure 1H-lane 9, Figure 2C-lane 8, Figure 2D-lane 6, and Figure 3C-lane 4) was claimed to have no activity but did exhibit endonuclease activity comparable to that of purified TS-SMG6TP-wt (Figure 1H-lane 5, Figure 2C-lane 2, Figure 2D-lane 2, and Figure 3C-lane 2). Thus, the enhanced endonucleolytic cleavage activities of TS-SMG6TP-wt (Figure 1H-lanes 11-13) or some of SMG6 mutants (Figure 2C-lanes 12-13, Figure 2D-lane 9) observed by adding Flag-Smg5p-wt may not have resulted from interactions of the SMG5-SMG6 PIN domains, but from independent activities of individual domains. In fact, purified Flag-SMG5—C1, R1, I3 (Figure 3C-lanes 5, 6, and 8) also exhibited significant cleavage activity. This raises the possibility that purified SMG5 fragments were contaminated by endo or exonucleases and question the validity of the conclusion “direct SMG5-SMG6 interactions stimulate endonucleolytic activity in vitro.”

Although we cannot exclude that traces of highly processive nucleases may be present in the protein preparations, we checked the purified samples for contaminants using mass spectrometry analysis, and did not detect a significant enrichment of any contaminant nuclease. Even in the case of traces of highly processive nucleases giving rise to background activity, we note that there is a drastic increase of the catalytic activity of TS-SMG6_{TP} WT in the presence of FLAG-SMG5_P WT, which is not found for the established catalytic mutant (D1353A) of TS-SMG6_{TP} in presence of FLAG-SMG5_P WT (e.g. compare Fig. 1H lanes 5, 13 and 15).

The in vitro nuclease assays we reported in the first submission were performed with a linear RNA, and as such we could not distinguish between endonucleolytic and exonucleolytic degradation of the substrate. We followed up on the Reviewer's comment, and in the revised version of the manuscript include in vitro nuclease assays performed with a circularized DNA-RNA-hybrid substrate to be able to distinguish between exonucleolytic and endonucleolytic activities (Extended Fig. 1G). Using this substrate, we observe some endonucleolytic activity for TS-SMG6_{TP} WT which is clearly enriched over the weak background activity of TS-SMG6_{TP} D1353A and FLAG-SMG5_P WT, both individually and together (Extended Data Fig. 1G, lane 2 compared with lanes 3, 4 and 6). Furthermore, we again observe the drastic stimulation of the endonucleolytic activity of TS-SMG6_{TP} WT in the presence of FLAG-SMG5_P WT, which even appears to extend to DNA bases (lane 5). These data thus indicate the presence of specific endonucleolytic activity. We also added this information to the revised manuscript and now also mention the background activity of the purified proteins for clarity.

In the manuscript we state (lines 138-149): "To confirm that the observed stimulatory effect of FLAG-SMG5_P on TS-SMG6_{TP} was indeed specific to the endonucleolytic activity of SMG6 and not the mediated by contaminating nucleases, we repeated the in vitro nuclease assays with a circularized substrate (Extended Data Fig. 1F), which had been generated by self-ligating a single-stranded DNA-RNA hybrid oligonucleotide with an internal Fluorescein label (5'-dT7-iFluorT-dT2-U30-3'). As with the linear substrate, we observed weak catalytic activity of the wild-type TS-SMG6_{TP} protein alone, which was strongly stimulated in presence of FLAG-SMG5_P (Extended Data Fig. 1G). Interestingly, the in vitro activity of TS-SMG6_{TP} alone also appeared to be expanded to the deoxyribose-phosphate backbone in presence of FLAG-SMG5_P, although the SMG6 PIN domain alone is unable to use single-stranded DNA as a substrate in vitro. We note that the residual activity observed with the catalytic mutant of TS-SMG6_{TP} or with of FLAG-SMG5_P alone are most likely background activities originating from contaminating nucleases in the protein preparations. "

2) To strengthen the conclusion that SMG5 and SMG6 form a composite PIN domain essential for NMD, the effects of different SMG5 and SMG6 mutants should be evaluated either in vitro or in vivo. Examples of the types of mutants that ought to be examined are those that specifically disrupt postulated protein-protein or RNA-protein interactions or postulated catalytic or RNA binding functions. These mutants, in the context of full-length proteins, should also be evaluated for in vivo NMD activities with bona fide NMD substrates, i.e., preferably not the transcripts currently analyzed in this manuscript including SRSF2 (Figures 1C, 2B, 3B, 4B, Extended Data Figure 4A), RBM3 (Extended Data Figures 1A, 2C, 3B) and SRSF6 (Extended Data Figures 1A, 2D, 3C, 4B). These transcripts all exhibit isoform switches to different extents and the increased accumulation of the PTC+ isoforms observed in NMD-deficient cells may only arise partially from direct effects of NMD loss. Analyses of bona fide NMD substrates in different SMG5 and

SMG6 mutants have the potential to detect different decay intermediates and thus facilitate functional assignments for different residues in both SMG5 and SMG6.

We agree that for transcripts undergoing isoform switching, changes in PTC-containing isoforms may over- or underestimate NMD activity. To address this concern, we performed probe-based qPCR assays for two bona fide NMD substrates representing distinct NMD-activating mechanisms: SRSF2, in which the PTC is generated by alternative splicing (AS-NMD), and ZFAS1, which harbors a premature termination codon and represents a classical PTC-NMD substrate.

Analysis of both substrates fully confirmed our original conclusions. Specifically, the same SMG5 and SMG6 mutants previously identified as non-functional also failed to restore NMD activity toward these bona fide NMD targets. Thus, this expanded analysis validates the inactive mutants and further supports the conclusion that SMG5 and SMG6 cooperate through a composite PIN-domain architecture to execute NMD.

In addition, we extended our mutant characterization beyond the previously used siRNA-based knockdown assays by re-evaluating the SMG5 and SMG6 mutant panel using a degron-based depletion system. These experiments yielded results that are fully consistent with our prior findings, while revealing more nuanced differences in the degree of NMD impairment among individual mutants. For SMG6, several mutants that appeared uniformly inactive under siRNA-mediated depletion now exhibit graded loss-of-function. For SMG5, the overall effects are slightly reduced in magnitude in the degron system, but the same mutants remain the most severely compromised.

Taken together, these new in vivo data (i) confirm the identity of inactive SMG5 and SMG6 mutants using bona fide NMD substrates, (ii) strengthen the functional assignment of specific residues and domains, and (iii) reinforce our model that SMG5 and SMG6 act cooperatively through a composite PIN-domain interface that is essential for proper NMD execution. These new results have been incorporated into the revised manuscript.

3) In the SMG5 and SMG6 function rescue experiments (Figures 3B and 2B), the integrated exogenous SMG5 or SMG6 mutant alleles are highly over-expressed compared to that from their respective endogenous alleles (Figures 3B and 2B, both lane1, bands invisible). The apparent loss of NMD function for some of the SMG5 and SMG6 mutants may thus originate from dominant -negative inhibition. Some of these mutants should be tested in WT cells for dominant-negative activity.

We tested the relevant SMG5 and SMG6 mutant alleles in degron cells without depleting SMG5 or SMG6 to directly assess possible dominant-negative effects. We performed RT-PCR analysis using the targets SRSF2 and SRSF6, as well as qPCR analysis of SRSF2 and ZFAS1.

For SMG5, none of the mutants displayed any dominant-negative inhibition of NMD (Panel A-C; Figure below). These data indicate that the failure of SMG5 mutants to rescue NMD reflects a true loss of function rather than dominant-negative interference.

For SMG6, in contrast, the same mutants that failed to rescue NMD activity did exhibit dominant-negative inhibition when expressed in SMG6 degron cells but without adding dTAGV-1 (Panel D-F; Figure below). As an additional control, we also expressed these mutants in wild

type HEK293 cells and likewise observed dominant-negative effects (Panel G-I; Figure below). These findings suggest that these SMG6 mutants can still be included into NMD complexes and replace endogenous SMG6, but because they lack catalytic activity or interaction with SMG5 they block execution of NMD. Importantly, the observed dominant-negative behavior provides reassurance that these mutants are not nonfunctional, but instead retain partial physiological activity sufficient for pathway engagement.

4) AlphaFold structure prediction is a powerful tool. To evaluate the prediction results, it will be helpful if the authors provided some confidence measures for the structure predictions presented in this manuscript including Figures 1D, 2A, 3A, and Extended Data Figures 1B, 1C, 2A, 2B, 3A).

***We agree that the confidence measures are important to assess the quality of the predictions.
We now provide the corresponding information (see answer to Reviewer 2, point 1).***